# GENERATIVE BAYESIAN OPTIMIZATION: GENERATIVE MODELS AS ACQUISITION FUNCTIONS

**Rafael Oliveira, Daniel M. Steinberg & Edwin V. Bonilla**
CSIRO, Sydney, Australia
{rafael.dossantosdeoliveira, dan.steinberg, edwin.bonilla}@csiro.au

## ABSTRACT

We present a general strategy for turning generative models into candidate solution samplers for batch Bayesian optimization (BO). The use of generative models for BO enables large batch scaling as generative sampling, optimization of non-continuous design spaces, and high-dimensional and combinatorial design. Inspired by the success of direct preference optimization (DPO), we show that one can train a generative model with noisy, simple utility values directly computed from observations to then form proposal distributions whose densities are proportional to the expected utility, i.e., BO's acquisition function values. Furthermore, this approach is generalizable beyond preference-based feedback to general types of reward signals and loss functions. This perspective avoids the construction of surrogate (regression or classification) models, common in previous methods that have used generative models for black-box optimization. Theoretically, we show that the generative models within the BO process follow a sequence of distributions which asymptotically approximate an optimal target under certain conditions. We also evaluate the performance through experiments on challenging optimization problems involving large batches in high dimensions.

## 1 INTRODUCTION

Bayesian optimization (BO) has been a successful approach to solve complex black-box optimization problems by making use of probabilistic surrogate models, such as a Gaussian processes (GPs) (Rasmussen & Williams, 2006), and their uncertainty estimates (Shahriari et al., 2016; Garnett, 2023). BO methods have been particularly useful in areas such as hyper-parameter tuning for machine learning algorithms (Snoek et al., 2012), material design (Frazier & Wang, 2016), and robot locomotion (Calandra et al., 2016). The core idea of BO is to apply a Bayesian decision-theoretic framework to make optimal choices by maximizing an expected utility criterion, also known as an acquisition function. The corresponding expectations are taken under a Bayesian posterior over the underlying objective function. Thus, the Bayesian model provides a principled way to account for the uncertainty inherent to the limited amount of data and the noisy observations.

In many applications such as simulated scenarios (Azimi et al., 2010), one is able to run multiple evaluations of the objective function in parallel, even though the simulations themselves might be expensive to run. Common BO approaches to these batch settings incrementally build a set of candidates by sampling "fantasy" observations from the probabilistic model and conditioning on them before selecting the next candidate in the batch (Wilson et al., 2018). Although near-optimal batches can be selected this way, this approach is not scalable to very large batches in high-dimensional spaces, such as problems in protein design (Stanton et al., 2022; Gruver et al., 2023).

One of the most promising alternatives to batch BO has been to train a generative model as a proposal distribution informed by the acquisition function and then sample a batch from the learned proposal (Brookes et al., 2019; Stanton et al., 2022; Gruver et al., 2023; Steinberg et al., 2025). This approach comes with several advantages. Firstly, given a trained generative model, sampling is usually inexpensive. Secondly, existing general-purpose generative models can be used and fine-tuned for the optimization task at hand. Lastly, sampling avoids estimating the global optimum of an acquisition function, which can be hard. However, existing generative approaches to black-box optimization usually rely on fitting a surrogate (regression or classification) model first then training a generative

model on top of it (Stanton et al., 2022; Gruver et al., 2023; Steinberg et al., 2025). This two-stage process compounds approximation errors from both models and can increase the computational cost significantly when compared to having a single model.

In this paper we present a general framework for learning generative models for batch Bayesian optimization tasks that requires a single model without the need for additional probabilistic regression or classification surrogates. Our approach for generative BO (GenBO) encodes general utility functions into training objectives for generative models directly. We focus on two cases, one where we train the model via a loss function for a reward model analogously to the direct preference optimization (DPO) formulation for large language models (Rafailov et al., 2023), and the second one where we train the generative model through divergence minimization, using utilities as part of sample weights. We present theoretical analyses on the convergence of approximations and empirical results on practical applications involving high-dimensional combinatorial optimization problems.

## 2 BACKGROUND

We consider the problem of estimating the global optimum of an objective function $f : \mathcal{X} \to \mathbb{R}$ as:

$$\mathbf{x}^* \in \underset{\mathbf{x} \in \mathcal{X}}{\operatorname{argmax}} f(\mathbf{x}), \tag{1}$$

where $f$ is an expensive-to-evaluate black-box function, i.e., $\nabla_{\mathbf{x}} f$ is unavailable. We can only observe $f(\mathbf{x})$ via noisy evaluations $y = f(\mathbf{x}) + \epsilon$, where $\epsilon$ is often assumed to be zero-mean Gaussian noise, though more general noise models can be handled by the framework we propose. We assume that the objective $f$ can be evaluated in parallel, and the algorithm is allowed to run up to $T \geq 1$ optimization rounds with a batch of $B$ query locations $\mathcal{B}_t := \{\mathbf{x}_{t,i}\}_{i=1}^B \subset \mathcal{X}$ per round.

**BO with regression models.** Typically BO assumes a Bayesian prior over $f$ (Garnett, 2023), often given by a Gaussian process (Rasmussen & Williams, 2006). Given a set of observations $\mathcal{D}_t$, corrupted by Gaussian noise $\epsilon \sim \mathcal{N}(0, \sigma_\epsilon^2)$, the Bayesian posterior distribution over $f$ given the data $\mathcal{D}_t$ is available in closed form as a GP with known mean and covariance functions (see Appendix A). BO then uses the model's posterior distribution to compute an acquisition function $a_t(\mathbf{x})$ mapping candidate points $\mathbf{x} \in \mathcal{X}$ to their expected utility value $\mathbb{E}[u(y)|\mathbf{x}, \mathcal{D}_t]$, where the utility function $u$ intuitively encodes how useful it is to collect a new observation at $\mathbf{x}$. Classical examples of expected utilities include the probability of improvement $a_t(\mathbf{x}) := p(y \geq \tau|\mathbf{x}, \mathcal{D}_t) = \mathbb{E}[\mathbb{I}[y \geq \tau]|\mathbf{x}, \mathcal{D}_t]$ and the expected improvement $a_t(\mathbf{x}) := \mathbb{E}[\max\{y - \tau, 0\}|\mathcal{D}_t]$. The next candidate is then chosen as:

$$\mathbf{x}_{t+1} \in \underset{\mathbf{x} \in \mathcal{X}}{\operatorname{argmax}} a_t(\mathbf{x}). \tag{2}$$

**Batch BO.** This strategy can be extended to the batch setting in a variety of ways (Garnett, 2023, §11.3). For instance, one can select the first batch point $\mathbf{x}_{t,1}$ by maximizing $a_t$ as above, and then select the next candidate as $\mathbf{x}_{t,2} \in \operatorname{argmax}_{\mathbf{x} \in \mathcal{X}} \mathbb{E}[u(y)|\mathbf{x}, \mathcal{D}_t \cup \{\mathbf{x}_{t,1}, \tilde{y}_{t,1}\}]$, where the expectation is over both $\tilde{y}_{t,1} \sim p(y|\mathbf{x}_{t,1}, \mathcal{D}_t)$ and $y \sim p(y|\mathbf{x}, \mathcal{D}_t \cup \{\mathbf{x}_{t,1}, \tilde{y}_{t,1}\})$, and iterate over this process until $B$ candidates have been selected for parallel evaluation. Although near optimal, evaluating this conditional expectation becomes quickly intractable as the batch size grows. Hence, one usually resorts to Monte Carlo approximations (Wilson et al., 2018). Other BO strategies allow for efficient optimization of the batch in parallel, such as information-theoretic acquisition functions (Takeno et al., 2020; Teufel et al., 2024), or even asynchronously (Kandasamy et al., 2018). However, scaling up to large batches in high-dimensional domains, especially involving combinatorial or mixed discrete-continuous search spaces, remains challenging (González-Duque et al., 2024).

**Active generation with classification models.** Instead of relying on a Bayesian surrogate model for $f$ and then computing an acquisition function $a$ on top of it, one can model $a$ directly, which is the main idea behind likelihood-free BO (Song et al., 2022). On this line, methods like variational search distributions (VSD, Steinberg et al., 2025) and batch BORE (Oliveira et al., 2022) learn a probabilistic classifier $\pi(\mathbf{x}) \approx p(y \geq \tau)$ in the original space, $\mathcal{X}$, based on improvement labels $z := \mathbb{I}[y \geq \tau]$ and then generate batches by approximately sampling $B$ candidates from $p(\mathbf{x}|y \geq \tau)$. The classifier can be learned by, e.g., minimizing the cross-entropy loss:

$$L_n(\pi) := -\sum_{i=1}^n z_i \log \pi(\mathbf{x}_i) + (1 - z_i) \log(1 - \pi(\mathbf{x}_i)). \tag{3}$$

Given a prior $p_0$ over $\mathcal{X}$ and the classifier $\pi_t$ that minimizes $L_{n_t}$ over the current $n_t := tB$ data points in $\mathcal{D}_t$, we can now learn a generative model approximating $p(\mathbf{x}|y \geq \tau, \mathcal{D}_t)$ as:

$$q_t \in \underset{q}{\operatorname{argmax}}\, \mathbb{E}_{\mathbf{x} \sim q}[\log \pi_t(\mathbf{x})] - \mathbb{D}_{\mathrm{KL}}(q||p_0)\,, \tag{4}$$

which corresponds to an evidence lower bound treating $\pi_t(\mathbf{x}) \approx p(y \geq \tau | \mathbf{x}, \mathcal{D}_t)$ as a likelihood.

**Direct preference optimization.** The process above for learning $q_t$ can be likened to the typical fine-tuning of large language models (LLMs) via reinforcement learning with human feedback (RLHF, Bai et al., 2022), which would normally involve training the LLM as an RL agent with a reward model $\rho$. In practice, we do not directly observe rewards, but have access to user preferences. Given a prompt's context $c$, corresponding to the RL state, let $\mathbf{x}^+, \mathbf{x}^- \sim q(\mathbf{x}|c)$ denote two answers generated by an LLM $q$, with $\mathbf{x}^+$ denoting the answer preferred by the user, and $\mathbf{x}^-$ the dispreferred one. Having a dataset $\mathcal{D}_n^+ := \{c_i, \mathbf{x}_i^+, \mathbf{x}_i^-\}_{i=1}^n$, one can then learn a reward function $\rho$ by minimizing the negative log-likelihood under a preference model, such as Bradley & Terry (1952):

$$L_n^+(\rho) := -\mathbb{E}_{(c,\mathbf{x}^+,\mathbf{x}^-) \sim \mathcal{D}_n^+}[\log \sigma(\rho(c, \mathbf{x}^+) - \rho(c, \mathbf{x}^-))]\,. \tag{5}$$

Having learned a reward model $\rho_n$, RLHF trains the LLM as to approximate an agent's optimal policy under $\rho_n$. Regularization based on the Kullback-Leibler (KL) divergence with respect to a reference model $q_{\mathrm{ref}}$ is further added to improve stability. The optimal generative model then solves:

$$q_n \in \underset{q}{\operatorname{argmax}}\, \mathbb{E}_{c \sim \mathcal{D}_n^+, \mathbf{x} \sim q(\mathbf{x}|c)}[\rho_n(c, \mathbf{x})] - \eta \mathbb{D}_{\mathrm{KL}}(q||q_{\mathrm{ref}})\,. \tag{6}$$

Direct preference optimization (DPO, Rafailov et al., 2023) removes the need for an explicit reward model by viewing the LLM itself through the lens of a reward model. It is not hard to show that, fixing a reward model $\rho$, the optimal solution to Equation 6 is given by:

$$q(\mathbf{x}|c) = \frac{1}{\zeta_\rho(c)} q_{\mathrm{ref}}(\mathbf{x}|c) \exp\left(\frac{1}{\eta}\rho(c, \mathbf{x})\right)\,, \tag{7}$$

where $\zeta_\rho(c) := \sum_{\mathbf{x}} q_{\mathrm{ref}}(\mathbf{x}|c) \exp(\eta^{-1}\rho(c, \mathbf{x}))$ is the partition function at the given context $c$. Although it is intractable to evaluate $\zeta_\rho$ in practice, DPO uses the fact that, in the Bradley-Terry model, the partition function-dependent terms cancel out. Note that the reward model $\rho$ can be expressed in terms of the optimal $q$ as:

$$\rho(c, \mathbf{x}) = \eta \log\left(\frac{q(\mathbf{x}|c)}{q_{\mathrm{ref}}(\mathbf{x}|c)}\right) + \eta \log \zeta_\rho(c)\,. \tag{8}$$

Applying the substitution above to the preference-based loss (5), we get:

$$L_{\mathrm{DPO}}(q) = -\mathbb{E}_{(c,\mathbf{x}^+,\mathbf{x}^-) \sim \mathcal{D}_n^+}\left[\log \sigma\left(\eta \log\left(\frac{q(\mathbf{x}^+|c)}{q_{\mathrm{ref}}(\mathbf{x}^+|c)}\right) - \eta \log\left(\frac{q(\mathbf{x}^-|c)}{q_{\mathrm{ref}}(\mathbf{x}^-|c)}\right)\right)\right]\,, \tag{9}$$

which eliminates the partition function $\zeta_\rho$ terms. Therefore, we can train the generative model $q$ directly with $L_{\mathrm{DPO}}$ without the need for an intermediate reward model. Such simplification to a single training loop cuts down the need for computational resources, eliminates a source of approximation errors (from learning $\rho$), and brings in theoretical guarantees from Bradley-Terry models (Shah et al., 2016; Bong & Rinaldo, 2022). The main question guiding our work is whether we can apply a similar technique to simplify the training of (arbitrary, not necessarily LLM) generative models for likelihood-free BO by removing the need for an intermediate surrogate model for $f$.

## 3 A GENERAL RECIPE FOR GENERATIVE BAYESIAN OPTIMIZATION

As seen in Section 2, using generative models for BO typically involves training a regression or classification model as an intermediate step to then train the candidate generator. The use of an intermediate model demands additional computational resources and brings in further sources of approximation errors which may hinder performance. Hence, we propose a framework to train the generative model directly from (noisy) observation values. The main idea is to train the model to approximate a target distribution proportional to BO's acquisition function and then use the learned generative model as a proposal for the next query locations. There are different approaches to do so, some of which have been previously explored in the literature for specific acquisition functions, such as the probability of improvement (Brookes et al., 2019; Steinberg et al., 2025) and upper confidence bound (Yun et al., 2025). However, we here focus on a general recipe to turn a generative model into a density following *any* acquisition function that can be expressed as an expected utility.

**Utility functions.** Consider a likelihood-free BO setting (Song et al., 2022), where we aim to directly learn an acquisition function $a_t : \mathcal{X} \to \mathbb{R}$ from available data $\mathcal{D}_t$. If our acquisition function takes the form of an expected utility:

$$a_t(\mathbf{x}) = \mathbb{E}[u_t(y) \mid \mathbf{x}, \mathcal{D}_t], \tag{10}$$

we can estimate $a_t$ from noisy utility data $\{\mathbf{x}_i, u_{t,i}\}_{i=1}^{n_t}$, where $u_{t,i} = u_t(y_i)$. As examples of utility functions, we have:

1. Probability of improvement (PI): $u_t(y) = \mathbb{I}[y \geq \tau_t]$;
2. Expected improvement (EI): $u_t(y) = \max(y - \tau_t, 0)$;
3. Soft expected improvement (sEI): $u_t(y) = \mathrm{softplus}(y - \tau_t)$;
4. Mean: $u_t(y) = y$;

given a threshold $\tau_t$ for improvement-based utilities, e.g., the largest observation value or a quantile of the empirical observations distribution (Tiao et al., 2021). A comprehensive summary of typical utility functions for BO can be found in Wilson et al. (2018). The ones listed above, however, can be directly expressed as a function of the observations. The soft-plus version of EI (sEI) is added as a smoother utility variant of EI which remains positive when $y \leq \tau$ (cf. Ament et al., 2023).

**BO with generative models.** As an illustrative example, consider the case of PI where $a(\mathbf{x}) = \mathbb{E}[\mathbb{I}[y \geq \tau] \mid \mathbf{x}] = p(y \geq \tau | \mathbf{x})$, which has been previously applied to train generative models for black-box optimization using reward surrogate models (Steinberg et al., 2025). Given a sampler for the conditional distribution $p(\mathbf{x}|y \geq \tau)$, by Bayes rule, we recover the original PI as:

$$a(\mathbf{x}) = p(y \geq \tau | \mathbf{x}) = \frac{p(\mathbf{x}|y \geq \tau)p(y \geq \tau)}{p_0(\mathbf{x})} \propto \frac{p(\mathbf{x}|y \geq \tau)}{p_0(\mathbf{x})}. \tag{11}$$

The prior $p_0$ is usually known, and it can be set as uninformative $p_0(\mathbf{x}) \propto 1$ or set to encode prior information about the optima. We then see that learning a generative model to approximate the posterior above is equivalent to learning a probabilistic classifier for the improvement event $y \geq \tau$, as in Song et al. (2022). Moreover, if we only have a probabilistic classifier approximating $p(y \geq \tau | \mathbf{x})$, we still need to select candidate points via optimization over the classification probabilities landscape, which can be highly non-convex presenting several local optima, as in the usual BO setting we choose $\mathbf{x}_{t+1}$ as the (global) maximizer of the acquisition function $a$. In contrast, a generative model provides us with a direct way to sample candidates $\mathbf{x} \sim p(\mathbf{x}|y \geq \tau)$ which will naturally concentrate in the regions of highest probability density, and therefore highest utility, according to the model. Finally, note that this same reasoning can be extended to any other non-negative expected utility function by training the generative model to approximate:

$$p_t^*(\mathbf{x}) \propto p_0(\mathbf{x})a_t(\mathbf{x}), \tag{12}$$

or similarly $p_t^*(\mathbf{x}) \propto p_0(\mathbf{x})\exp a_t(\mathbf{x})$, which allows for utilities that might take negative values. Thus, GenBO admits a generalized Bayesian interpretation as direct inference over the optimum location $\mathbf{x}^*$ rather than $f$, with $p_0$ serving as a prior over $\mathbf{x}^*$ and the acquisition value $a_t(\mathbf{x})$ as a utility-based pseudo-likelihood factor (cf. Knoblauch et al., 2022; Souza et al., 2021).

**Overview.** Let $\mathcal{Q} \subset \mathcal{P}(\mathcal{X})$ be a learnable family of probability distributions over a domain $\mathcal{X}$. We consider general loss functions of the form:

$$L_t(q) := \lambda_t R_t(q) + \sum_{i=1}^{n_t} \ell_i(q), \tag{13}$$

where $\ell_i$ are individual losses over points $\mathbf{x}_i \in \mathcal{X}$ or pairs of points $\mathbf{x}_{i,1}, \mathbf{x}_{i,2} \in \mathcal{X}$ and their corresponding utility values, $\lambda_t \geq 0$ is an optional regularization factor, and $R_t : \mathcal{Q} \to [0, \infty)$ is a complexity penalty function. The algorithm starts by learning a proposal from available data $\mathcal{D}_{t-1}$:

$$q_{t-1} \in \underset{q \in \mathcal{Q}}{\arg\min}\, L_{t-1}(q). \tag{14}$$

A batch $\mathcal{B}_t := \{\mathbf{x}_{t,i}\}_{i=1}^B$ is sampled from the learned proposal $q_{t-1}$. We evaluate the utilities $u_t(y_{t,i})$ with the collected observations $y_{t,i} \sim p(y|\mathbf{x}_{t,i})$, for $i \in \{1, \ldots, B\}$, and repeat the cycle up to a given number of iterations $T \in \mathbb{N}$. This process is summarized in Algorithm 1. In the following, we describe strategies to formulate general losses $\ell_i$ to learn acquisition functions and how to ensure that the sequence of batches $\{\mathcal{B}_t\}_{t=1}^\infty$ asymptotically concentrates at the global optima.

---
**Algorithm 1:** Generative BO
---
**Input:** Domain $\mathcal{X}$, initial data $\mathcal{D}_0$
**for** $t \in \{1, \ldots, T\}$ **do**
$\quad$ $q_{t-1} \in \operatorname{argmin}_{q \in \mathcal{Q}} L_{t-1}(q)$ $\qquad$ // Fit proposal distribution
$\quad$ $\mathcal{B}_t = \{\mathbf{x}_{t,i}\}_{i=1}^B \sim q_{t-1}$ $\qquad$ // Sample batch
$\quad$ $y_{t,i} \leftarrow f(\mathbf{x}_{t,i}) + \epsilon_{t,i}, \text{for } i \in \{1, \ldots, B\}$ $\qquad$ // Collect observations
$\quad$ $\mathcal{D}_t = \mathcal{D}_{t-1} \cup \{\mathbf{x}_{t,i}, y_{t,i}\}_{i=1}^B$ $\qquad$ // Update data
---

### 3.1 PREFERENCE-BASED LEARNING

We first aim to apply a similar reparameterization trick to the one in DPO to simplify generative BO methods. Note that, for a general classification loss, such as the one in Equation 3, it is not possible to eliminate the partition function resulting from a DPO-like reparameterization without resorting to approximations, which might change the learned model. Hence, we need a pairwise-contrastive training objective.

**Preference loss.** To apply a preference-based loss, we can train a model to predict preferential directions of the acquisition function. Assume we have a dataset $\mathcal{D}_n^u := \{\mathbf{x}_i, u_i\}_{i=1}^n$ with $n$ evaluations of a given utility function $u : \mathbb{R} \to \mathbb{R}$. We may reorganize the data into pairs of inputs and corresponding utility values $\{\mathbf{x}_{i,1}, \mathbf{x}_{i,2}, u_{i,1}, u_{i,2}\}_{i=1}^{n/2}$, where $u_{i,j} := u(y_{i,j})$, for $j \in \{1, 2\}$, and train a generative model $q$ using the Bradley-Terry preference loss from DPO with, for $i \in \{1, \ldots, n/2\}$:

$$\ell_i^{\mathrm{PL}}(q) := \ell_i^{\mathrm{PL}}(q, \Delta u_i) := -\log \sigma \left( \eta \operatorname{sign}(\Delta u_i) \left( \log \left( \frac{q(\mathbf{x}_{i,1})}{p_0(\mathbf{x}_{i,1})} \right) - \log \left( \frac{q(\mathbf{x}_{i,2})}{p_0(\mathbf{x}_{i,2})} \right) \right) \right), \quad (15)$$

where $\Delta u_i := u_{i,1} - u_{i,2}$, as in the DPO formulation, $\eta > 0$ is a (optional) temperature parameter and the prior $p_0$ can be given by a reference model, either pre-trained or derived from expert knowledge about feasible solutions to the optimization problem (1). Similar to Rafailov et al. (2023), the learned generative model is seeking to approximate:

$$p_u^*(\mathbf{x}) := \frac{1}{\zeta_u} p_0(\mathbf{x}) \exp \left( \frac{1}{\eta} \mathbb{E}[u(y)|\mathbf{x}] \right), \quad (16)$$

where $\zeta_u$ is the normalization factor.

**Robust preference loss.** As shown in Chowdhury et al. (2024), the original DPO loss is not robust to preference noise. As in BO, one usually only observes noisy evaluations of the objective function, utility values directly derived from the observation values will also be noisy and correspondingly the sign of their differences as well. Namely, assume there is a small $p_{\mathrm{flip}} \in (0, 1/2)$ probability of the preference directions being flipped w.r.t. the sign of the true expected utility:

$$\mathbb{P}\left[\operatorname{sign}(u_{i,1} - u_{i,2}) = \operatorname{sign}(\mathbb{E}[u_{i,2}|\mathbf{x}_{i,2}] - \mathbb{E}[u_{i,1}|\mathbf{x}_{i,1}])\right] = p_{\mathrm{flip}}. \quad (17)$$

Chowdhury et al. (2024) showed that the original DPO preference loss is biased in this noisy case, and proposed a robust version of the DPO loss to address this issue as:

$$\ell_i^{\mathrm{rPL}}(q) := \frac{(1 - p_{\mathrm{flip}}) \ell_{\mathrm{PL}}(q, \Delta u_i) - p_{\mathrm{flip}} \ell_i^{\mathrm{PL}}(q, -\Delta u_i)}{1 - 2p_{\mathrm{flip}}}, \quad (18)$$

which yields the robust preference loss (rPL). It follows that the loss function above is unbiased and robust to observation noise under mild assumptions (Chowdhury et al., 2024).

### 3.2 DIVERGENCE-BASED LEARNING

A disadvantage of DPO-based losses when applied to BO is that they only take the signs of the pairwise utility differences into account, discarding the remaining information contained in the magnitude of the utilities. A simpler approach is to train the generative model $q$ to match $p_u^*$ directly.

**Forward KL.** If we formulate the target distribution as $p_u^*(\mathbf{x}) \propto p_0(\mathbf{x})a(\mathbf{x})$, the forward Kullback-Leibler (KL) divergence of the proposal w.r.t. the target is given by:

$$\mathbb{D}_{\mathrm{KL}}(p_u^*||q) = \mathbb{E}_{\mathbf{x} \sim p_u^*}[\log p_u^*(\mathbf{x}) - \log q(\mathbf{x})]. \tag{19}$$

As we do not have samples from $p_u^*$, at each iteration $t$ the algorithm generates samples from the current best approximation $\mathcal{B}_t := \{\mathbf{x}_{t,i}\}_{i=1}^B \sim q_{t-1}$. An unbiased training objective can then be formulated as:

$$\ell_i^{\mathrm{fKL}}(q) = -\frac{p_0(\mathbf{x}_i)}{q_{i-1}(\mathbf{x}_i)} u(y_i) \log q(\mathbf{x}_i), \tag{20}$$

which we write in a condensed form to avoid notation clutter where $q_{i-1}$ denotes the proposal density used to sample $\mathbf{x}_i$ under the re-indexed sequence of observations. The objective above is unbiased and its global optimum can be shown to converge to $p_u^*$ by an application of standard results from the adaptive importance sampling literature (Delyon & Portier, 2018). A simpler version of this training objective was derived for CbAS (Brookes et al., 2019) using only the last batch for training, which would allow for convergence only as the batch size goes to infinity $B \to \infty$. Furthermore, as we discuss in our analysis, convergence to $p_u^*$ is not sufficient to ensure convergence to the global optima of the objective function $f$.

**Balanced forward KL.** As utilities like those of PI and EI can evaluate to 0 at the points where $y < \tau$ was observed, with $\tau$ corresponding to an improvement threshold, every point below the threshold will not be penalized by the loss function. As a result, the model may keep high probability densities in regions of low utility. To prevent this, we may use an alternative formulation of the forward KL which comes from the definition of Bregman divergences with the convex function $u \mapsto u \log u$, yielding a loss:

$$\ell_i^{\mathrm{bfKL}}(q) = -\frac{p_0(\mathbf{x}_i)}{q_{i-1}(\mathbf{x}_i)} u(y_i) \log q(\mathbf{x}_i) + \frac{q(\mathbf{x}_i)}{q_{i-1}(\mathbf{x}_i)}. \tag{21}$$

We defer the details of the derivation to the appendix. Although the additional $q(\mathbf{x})$ only contributes to a constant term when integrated over, for finite-sample approximations, it contributes to a soft penalty on points where it is observed that $u(y) = 0$.

### 3.3 GENERALIZATIONS

In general, we can extend the above framework to use a proper scoring rule $S : \mathcal{P}(\mathcal{X}) \times \mathcal{X} \to \mathbb{R}$ (Gneiting & Raftery, 2007) other than the log loss. We can then optimize $q$ using losses of the form:

$$\ell_i^S(q) = -\frac{p_0(\mathbf{x}_i)}{q_{i-1}(\mathbf{x}_i)} u(y_i) S(q, \mathbf{x}_i). \tag{22}$$

Although we leave the exploration of this formulation for future work, it is essentially compatible with our proposed theoretical framework and readily extensible to generative models that may not have densities available in closed form, such as diffusion and flow matching (Lipman et al., 2024), which still provide flexible probabilistic models.

## 4 THEORETICAL ANALYSIS

In this section, we present a theoretical analysis for the algorithm's approximation of the utility-based target distribution $p_u^*$ and its performance in regards to the global optimization problem (1). We consider parametric generative models $q_\theta$ with a given parameter space $\Theta \subset \mathbb{R}^M$. For the purpose of our analysis, we will assume that models can be described as $q_\theta(\mathbf{x}) = \exp g_\theta(\mathbf{x})$, which is possible whenever probability densities are strictly positive $q_\theta(\mathbf{x}) > 0$. To accommodate for both the pairwise preference-based losses and the point-based divergence approximations, we introduce the following notation for the loss function:

$$L_n(g_\theta) = R_n(g_\theta) + \sum_{i=1}^n w_i \ell(m_i(g_\theta), z_i), \tag{23}$$

where $m_i(g_\theta)$ corresponds to the $i$th model evaluation with, e.g., $m_i(g_\theta) := g_\theta(\mathbf{x}_i) = \log q_\theta(\mathbf{x}_i)$ for KL, and $m_i(g_\theta) := g_\theta(\mathbf{x}_{i,1}) - g_\theta(\mathbf{x}_{i,2})$ for preference-based losses, $z_i$ encodes the dependence on utility values with $z_i := u(y_i)$ for KL and $z_i := \mathrm{sign}(u_{i,1} - u_{i,2})$ for DPO losses, and $w_i$ are potential importance weights.

**Regularity assumptions.** We make a few mild regularity assumptions about the problem setting and the model. Firstly, for the analysis, we assume that the models $g_\theta$ lie in a reproducing kernel Hilbert space (RKHS) $\mathcal{H}_k$ shared with the true log density $g_*$, which is such that $p_u^*(\mathbf{x}) = \exp g_*(\mathbf{x})$. This assumption does not impose that the proposals are kernel models, but simply that they lie in Hilbert space of functions with well defined point evaluations. The domain $\mathcal{X}$ is assumed to be a compact metric space, with main results specialized for the finite discrete setting, i.e., $|\mathcal{X}| < \infty$. The individual losses $\ell : \mathbb{R} \times \mathbb{R} \to \mathbb{R}$ are strictly convex and twice differentiable w.r.t. their first argument. We also assume that the regularizer $R_n$ is strongly convex and twice differentiable. The rest of our assumptions and proofs are presented and discussed in more detail in Appendix B.

**Lemma 1.** *Let assumptions A2 to A4 be satisfied. Then,*

$$\frac{1}{2}\|g - g_n\|_{H_n}^2 \le L_n(g) - L_n(g_n) \le \frac{1}{2}\|\nabla L_n(g)\|_{H_n^{-1}}^2 \,,$$

*where $H_n : \mathcal{H}_k \to \mathcal{H}_k$ is an operator-valued lower bound on the Hessian of the loss $L_n$:*

$$\forall g \in \mathcal{H}_k, \quad \nabla^2 L_n(g) \succeq H_n := \lambda_n I + \alpha_\ell b_w \sum_{i=1}^n m_i \otimes m_i \,.$$

*Remark* 1. The result in Lemma 1 automatically ensures that the loss functional $L_n$ is strongly convex over the function space, as $\nabla^2 L_n(g) \succeq H_n \succeq \lambda_n I \succ 0$, for all $g \in \mathcal{H}_k$, and therefore has a unique minimizer at $g_n$, defined in Section B.3. Note, however, that the same cannot be implied about $L_n(g_\theta)$ over $\Theta$ based on this result alone, since the mapping $\theta \mapsto g(\cdot, \theta)$ may not be linear.

**Theorem 1.** *Let assumptions A1 to A5 hold. Then, given any $\delta \in (0,1)$,*

$$\forall n \in \mathbb{N}, \quad |\langle m, g_* \rangle_k - \langle m, g_{\theta_n} \rangle_k| \le \beta_n(\delta)(\|m\|_{H_n^{-1}} + 2|\langle m, \iota \rangle_k| v_n(\bar{q}_n)), \quad \forall m \in \mathcal{H}_k,$$

*which holds with probability at least $1 - \delta$, where $\beta_n(\delta)$ is given by Lemma 5, where $\iota(\mathbf{x}) = 1$ denotes the unit constant function, $v_n(q) := \mathbb{E}_{\mathbf{x} \sim q}\left[\|\phi(\mathbf{x})\|_{H_n^{-1}}\right]$, and $\bar{q}_n := \frac{p_u^* + q_n}{2}$.*

The result above shows that the approximation error for the optimal parameter $\theta_n$ concentrates similarly to that of a kernel method, even though we do not require the model to be a kernel machine. In addition, the term $\|m\|_{H_n^{-1}}$ is associated with the predictive variance of a Gaussian process model, which can be shown to converge to zero if, e.g., $\inf_{\mathbf{x} \in \mathcal{X}} q_\theta(\mathbf{x}) \ge b_q > 0$, for all $\theta \in \Theta$ (see Lemma 4 in the appendix). Alternatively, one can follow a KL-focused analysis as in Oliveira et al. (2021) to bound $\mathbb{D}_{\mathrm{KL}}(q_{\theta_n} \| q_*)$ via the information gain, which bounds the growth of predictive uncertainty terms $\|m_n\|_{H_{n-1}^{-1}}^2$ appearing in Theorem 1. The asymptotic rate of $\beta_n$ is discussed in Remark 4.

**Optimality.** Theorem 1 allows us to establish that the model converges to the target $g_*$ associated with the target distribution $p_u^*$ for a given utility function $u$, provided $\|m\|_{H_n^{-1}}$ vanishes sufficiently fast. However, convergence to the target distribution alone does not ensure optimality of the samples $\mathbf{x} \sim q_t$. The latter is possible by applying results from reward-weighted regression, which shows that training a proposal to maximize $\mathbb{E}_{y \sim p(y|\mathbf{x}), \mathbf{x} \sim q_{t-1}}[u(y) \log q(\mathbf{x})]$ yields a sequence of increasing expected rewards $\mathbb{E}[u(y_t)] \le \mathbb{E}[u(y_{t+1})] \le \dots$ (Štrupl et al., 2022, Thm. 4.1). If the maximizer of the sequence of expected utilities converges to the maximizer of the objective function $f$, then the generative BO proposals will concentrate at $f$'s global optima. Therefore, for KL-based loss functions, one may drop the proposal densities in the importance sampling weights $1/q_{i-1}(\mathbf{x}_i)$ to promote this posterior concentration phenomenon, as corroborated by our experimental findings, which generally did not include importance weights. This same concentration of the learned target distribution should also occur with the preference-based loss functions due to the absence of importance-sampling weights. If these targets concentrate on the optimizer set, simple regret may vanish, whereas sublinear cumulative regret requires rate control that we leave for future work.

## 5 RELATED WORK

Using generative models for online-optimization is becoming an increasingly popular strategy for optimization in discrete, mixed discrete-continuous or high-dimensional design spaces where classical BO is limited. The following discusses other works applying generative models to BO settings and contrasts them with the reward-model-free active generation framework we propose.

**Latent-space BO.**  In latent-space BO (LSBO) methods for high-dimensional problems (Gómez-Bombarelli et al., 2018; Stanton et al., 2022; Gruver et al., 2023), one learns a probabilistic representation of a (usually lower-dimensional) manifold of the data jointly with $f$, and performs BO in that space, projecting query points back to the original space at evaluation time. This technique has led to numerous BO methods for high-dimensional and discrete-space optimization (Gómez-Bombarelli et al., 2018; Gruver et al., 2023; González-Duque et al., 2024). Learning this latent space can, however, cause complications. LSBO can suffer poor sample efficiency if the latent space is learned from the initial training set and then fixed (Tripp et al., 2020). Or poor performance can arise from reconstruction errors between the latent and observation space (Lee et al., 2025). GenBO and other methods like VSD do not suffer from these issues as all inference is done in the observation space. Lastly, despite recent advances in the field (Chu et al., 2024; Lee et al., 2025; Moss et al., 2025), to our knowledge, LaMBO-2 remains state-of-the-art in LSBO for *long* sequences, like proteins.

**Diffusion for BBO.**  There has been recent progress in adapting denoising diffusion models to black-box optimization (BBO) tasks, often by learning a model that can be conditioned on observation values, given a dataset of evaluations (Krishnamoorthy et al., 2023). Other approaches involve guiding the diffusion process by a given utility function derived from a regression model (Gruver et al., 2023; Yun et al., 2025). Note, however, that such methodologies are specific to diffusion, whereas we focus on a general approach that can be applied to arbitrary generative models.

**LLMs and BO.**  Recent work has begun to integrate large language models (LLMs) into BO pipelines, primarily to inject prior knowledge, improve cold-start performance, or offload certain design decisions to a learned policy. Several studies use LLMs as contextual priors over the design space: for example, guiding initialization or proposal generation by leveraging natural-language domain knowledge (Liu et al., 2024), or selecting acquisition functions adaptively via an LLM-driven controller (Aglietti et al., 2025). Other work treats BO as a test-time search tool that an LLM can call to refine or validate its own proposals during inference (Agarwal et al., 2025). Most relevant to our setting is a recent reward-model-free approach for protein engineering (Chen et al., 2025), which uses LLM preference modeling, akin to DPO, to steer search without an explicit surrogate. This shares the reward-model-free philosophy of GenBO, but differs fundamentally in relying on a general-purpose LLM, whereas GenBO provides a framework for task-specific generative black-box optimization problems with no language interface or pretrained reward structure.

## 6 EXPERIMENTS

We evaluate several variants of generative BO (GenBO) on a number of challenging sequence optimization tasks against popular and strong baselines, including CbAS (Brookes et al., 2019), VSD (Steinberg et al., 2025), and LaMBO-2 (Gruver et al., 2023), besides trivial baselines, random mutations and a genetic algorithm (GA) implemented in POLI (González-Duque et al., 2024). As performance measures, we assess the simple regret, $r_t := f(\mathbf{x}^*) - \max_{i \leq n_t} f(\mathbf{x}_i)$, and the cumulative maximum, $\max_{i \leq n_t} f(\mathbf{x}_i)$, where $n_t := |\mathcal{D}_t|$ is the number of function evaluations up to round $t$. In legend boxes, algorithms are sorted in descending order of final average regret. Shaded areas correspond to $\pm 1$ standard deviation across five different random seeds. Appendix C presents further details about experiment settings and ablation studies. Table 4 and 5 summarize final results.

### 6.1 TEXT OPTIMIZATION

As a first experiment, we wish to optimize a short sequence (5 letters) to minimize the edit distance to the sequence ALOHA, which is implemented as a POLI black-box (González-Duque et al., 2024). Here $\mathcal{X} = \mathcal{V}^M$ where $\mathcal{V}$ is the English alphabet, and $M$ is sequence length. Even though this sequence is relatively short, still $|\mathcal{X}| = |\mathcal{V}|^M > 11.8$ million elements. We increase the difficulty by only allowing $|\mathcal{D}_0| = 64$ where the minimum edit distance is 4, $B = 8$, and $T = 10$. We compare GenBO to the classifier guided VSD (Steinberg et al., 2025) and CbAS (Brookes et al., 2019), and to a simple greedy baseline that applies (3) random mutations to its best candidates per-round (González-Duque et al., 2024). For GenBO, VSD and CbAS we use a simple mean-field (independent) categorical proposal distribution, $q$, and a uniform prior, $p_0$. VSD and CbAS use a simple embedding and 1-hidden layer MLP classifier for estimating PI. We also varied the threshold $\tau$ annealing schedule. Architectural details and other experimental specifics are given in Section C.1.

Results are summarized in Figure 1a. We can see that the random baseline is not able to make much headway and CbAS under-performs due to its limited use of data (last batch only) in retraining. For this experiment, GenBO with the robust preference loss (rPL) and EI-based utilities showed the quickest improvements, whereas PI is able to reach the exact optimum at the end, with VSD eventually also achieving good performance. In Figure 5 (appendix), we present an ablation study on the threshold $\tau_t$ annealing scheme we used to balance the exploration-exploitation trade-off for GenBO and PI-based baselines (VSD and CbAS). The plots reveal that this problem generally favors a more exploitative approach by concentrating on higher quantiles of the observations marginal distribution. GenBO was, however, relatively less sensitive to the choice of annealing scheme, as long as the final percentile was set anywhere above 90%, whereas VSD required a generally sharper rise to above the 95% quantile towards the end of the optimization process, favoring original settings suggested by Steinberg et al. (2025). We also find that in this problem the use of a pre-trained informative prior $p_0$ may not bring significant performance advantage, as GenBO variants with no prior (i.e., $p_0 \propto 1$) performed best. Lastly, we also highlight significant improvements in run time for GenBO, making it on average 3 times faster than VSD (see Table 6 in the appendix) for not needing to fit an intermediate surrogate model.

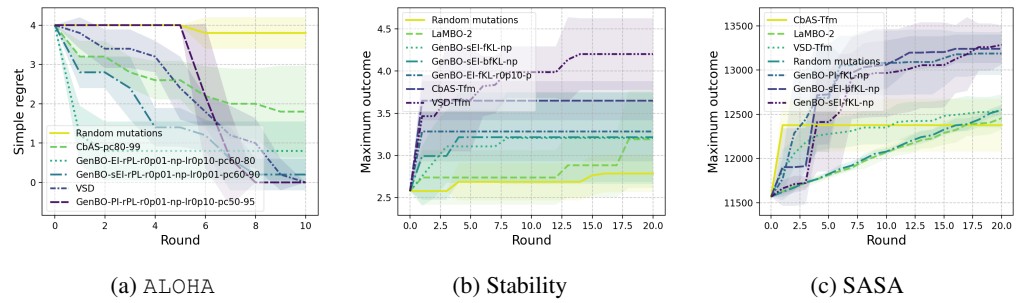

(a) ALOHA      (b) Stability      (c) SASA

Figure 1: Performance of baseline black box optimizers and GenBO variants on the (a) ALOHA, (b) stability, and (c) solvent accessible surface area optimization problems.

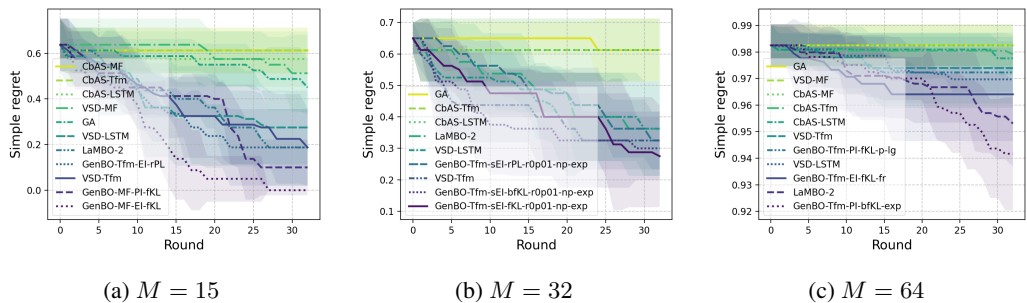

(a) $M = 15$      (b) $M = 32$      (c) $M = 64$

Figure 2: Simple regret of the baseline black box optimizers and the GenBO variants on the Ehrlich closed-form test function protein design task for varying sequence lengths, $M$.

## 6.2 PROTEIN DESIGN

We now consider three protein sequence design tasks where $|\mathcal{V}| = 20$ and we have varying $M$. We again use VSD, CbAS and random mutation as baselines, and add to them the guided diffusion based LaMBO-2 (Gruver et al., 2023). GenBO, VSD and CbAS all share the same generative backbone, which is the causal transformer used in Steinberg et al. (2025); VSD and CbAS also use the same CNN-classifier guide used in that work. We present additional architectural information, and additional experimental details in Section C.2. We use the black-box implementations in POLI for these tasks, and POLI-BASELINES implementations of the random and LaMBO-2 baselines.

The first task we consider is optimization of the Ehrlich functions introduced by Stanton et al. (2024). These are challenging biologically inspired parametric closed-form functions that explicitly

simulate nonlinear (epistatic) effects of sequence on outcome. The outcomes are $y \in \{-1\} \cup [0, 1]$ where $-1$ is reserved for infeasible sequences. We use the same protocol as in Steinberg et al. (2025), where we optimize sequences of length $M = \{15, 32, 64\}$ all with motif lengths of 4, and $|\mathcal{D}_0| = 128$, $T = 32$ and $B = 128$. The results are summarized in Figure 2. We again see that GenBO variants are able to outperform or match the performance of baselines, with KL-based losses yielding the best performance. In higher dimensions with the longest sequence setting, the benefits of the balanced forward KL loss, with its density minimization effect in areas of lower utility, are more evident. In addition, we note that exponential regularization, corresponding to assuming an exponential dot-product kernel for the RKHS feature space of the model (see Remark 2), allowed for the best performance in higher dimensions. Lastly, in Figure 6 (appendix), we present an ablation study on the batch size setting $B$, showing monotonic improvements, especially for large $B \geq 32$.

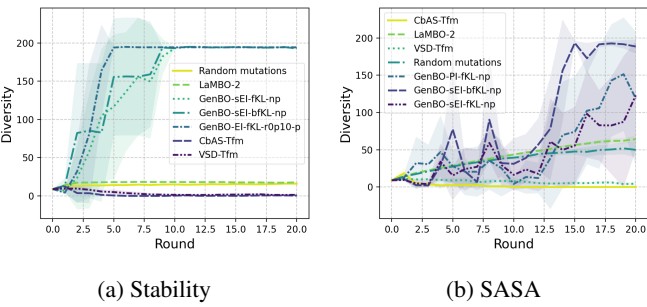

(a) Stability    (b) SASA

Figure 3: Batch diversity scores per round on the FoldX protein optimization tasks.

For our final set of experiments we present two real protein optimization tasks. These experiments have been adapted from Stanton et al. (2022) where the aims are to maximize the stability and solvent accessible surface area (SASA) of the proteins, respectively. The black-box is the FoldX molecular simulation software (Schymkowitz et al., 2005), and is wrapped by POLI (González-Duque et al., 2024). We chose the mRouge red fluorescent protein ($M = 228$) as the base protein for the tasks. Both tasks were run for $T = 20$ rounds with a batch size of $B = 64$ and an initial training set of $|\mathcal{D}_0| = 88$ as a subset from Stanton et al. (2022). Results are summarized in Figure 1b for stability and Figure 1c for SASA. All variants of GenBO find the stability task challenging, along with the LaMBO-2 and random baselines. CbAS and especially VSD are better able to stabilize this protein. As shown by diversity scores in Figure 3a, which we measure by averaging the Levenshtein distance across the batch in the same way as Steinberg et al. (2025), algorithmic baselines with the lowest diversity yielded top performance, indicating that pure exploitation from around the starting dataset led to the highest outcomes. However, most variants of GenBO far outperform the baselines on the SASA task, and much more rapidly. We believe this task favors extrapolation away from the prior, due to the high performance of GenBO variants with uninformative prior. In contrast to the stability, the diversity scores show that increasing exploration led to better outcomes for SASA (Figure 3b).

## 7    CONCLUSION

This work introduces Generative Bayesian Optimization (GenBO), a unifying framework that turns any generative model into a sampler whose density tracks BO acquisition functions. We have shown that loss functions over generative models, such as DPO and KL divergences, can be applied to directly learn samplers for batch BO. By eliminating intermediate regression or classification surrogates, GenBO reduces approximation error, simplifies the pipeline to learning just a single generative model, and scales naturally to large batches and high-dimensional or combinatorial design spaces. Theoretical results show convergence to the target distribution, and experiments on text optimization and protein design tasks demonstrate competitive performance with more complex surrogate-guided baselines. A few challenges remain. For some variants, GenBO requires choosing and fixing the prior before optimization, and its performance depends on sensible settings of utility and temperature parameters, whose theory could be further explored. Another avenue is the adaptation to acquisition strategies not expressible as expected utilities, such as Thompson sampling and upper confidence bound. Despite these caveats, GenBO's minimal moving parts and principled acquisition-driven training mark a simpler and more scalable alternative to multi-stage guided generation methods.

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

# A  GAUSSIAN PROCESSES FOR BO

Assume a Gaussian process prior over $f$, e.g., $f \sim \mathcal{GP}(0, k)$, where $k : \mathcal{X} \times \mathcal{X} \to \mathbb{R}$ is a positive-definite kernel (Rasmussen & Williams, 2006). Then, given a set of observations $\mathcal{D}_n := \{\mathbf{x}_i, y_i\}_{i=1}^n$, corrupted by Gaussian noise $\epsilon \sim \mathcal{N}(0, \sigma_\epsilon^2)$, the posterior $f|\mathcal{D}_n \sim \mathcal{GP}(\hat{f}_n, k_n)$ is available in closed form with mean and covariance function given by:

$$\hat{f}_n(\mathbf{x}) := \mathbf{k}_n(\mathbf{x})^\mathsf{T}(\mathbf{K}_n + \sigma_\epsilon^2 \mathbf{I})^{-1}\mathbf{y}_n \tag{24}$$

$$k_n(\mathbf{x}, \mathbf{x}') := k(\mathbf{x}, \mathbf{x}') - \mathbf{k}_n(\mathbf{x})^\mathsf{T}(\mathbf{K}_n + \sigma_\epsilon^2 \mathbf{I})^{-1}\mathbf{k}_n(\mathbf{x}') \tag{25}$$

$$\sigma_n^2(\mathbf{x}) := k_n(\mathbf{x}, \mathbf{x}), \tag{26}$$

where $\mathbf{k}_n(\mathbf{x}) := [k(\mathbf{x}, \mathbf{x}_i)]_{i=1}^n \in \mathbb{R}^n$, $\mathbf{K}_n := [k(\mathbf{x}_i, \mathbf{x}_j)]_{i,j=1}^n \in \mathbb{R}^{n \times n}$, $\mathbf{y}_n := [y_i]_{i=1}^n \in \mathbb{R}^n$, for $\mathbf{x}, \mathbf{x}' \in \mathcal{X}$. With these closed-form expressions, GP models allow BO algorithms to quantify uncertainty and assess expected utilities of their decisions. However, note that, due to matrix inversions, exact GP inference incurs a computational cost of $\mathcal{O}(n^3)$. Hence, one often has to resort to low-rank approximations to make GP predictions tractable in cases involving large amounts of data, such as batch evaluations with large batch size. Alternatively, one may completely discard the GP models and use other surrogates, such as neural networks, and there has been an increasing literature on how to reliably quantify uncertainty for BO when using these models (Li et al., 2024).

# B  LEARNING PARAMETRIC MODELS WITH RKHS CONVEX LOSSES

In this section, we consider the general problem of learning a function $g_*$ with a parametric model $g : \mathcal{X} \times \Theta \to \mathbb{R}$, where the parameter space $\Theta$ is an arbitrary finite-dimensional vector space. Most existing results in the Bayesian optimization and bandits literature for learning these models from inherently dependent data are only valid for linear models or kernel machines. As we will consider arbitrary generative models, we need to derive convergence results applicable to a wider class of models, accommodating popular modern frameworks. To do so, we will assume that there exists a reproducing kernel Hilbert space (RKHS) containing the models and the target log-density $g_* = \log p_u^*$ for a given (fixed) utility function $u$.

## B.1  MAIN DEFINITIONS

We will need the following definitions to state our main assumptions and results.

**Definition 1** (RKHS). *Let $\mathcal{X}$ be a non-empty set. A Hilbert space $\mathcal{H}$ of real-valued functions over $\mathcal{X}$ is called a reproducing kernel Hilbert space if function evaluations are bounded, that is:*

$$\forall \mathbf{x} \in \mathcal{X}, \quad \exists c_\mathbf{x} < \infty : \quad \forall h \in \mathcal{H}, \quad h(\mathbf{x}) \leq c_\mathbf{x}\|h\|_\mathcal{H}.$$

Any Hilbert space $\mathcal{H}$ satisfying the condition above is associated with a unique positive-semidefinite kernel $k : \mathcal{X} \times \mathcal{X} \to \mathbb{R}$ with the reproducing property (Schölkopf & Smola, 2002):

$$\forall h \in \mathcal{H} \qquad h(\mathbf{x}) = \langle h, k(\cdot, \mathbf{x})\rangle_\mathcal{H}, \qquad \forall \mathbf{x} \in \mathcal{X}, \tag{27}$$

where $\langle \cdot, \cdot \rangle_\mathcal{H}$ denotes the inner product in $\mathcal{H}$, which also defines the norm $\|h\|_\mathcal{H} = \sqrt{\langle h, h\rangle}_\mathcal{H}$, for $h \in \mathcal{H}$. In fact, it can be shown that $\mathcal{H}$ is spanned by $\{k(\cdot, \mathbf{x})\}_{\mathbf{x} \in \mathcal{X}}$ (Steinwart & Christmann, 2008, Thm. 4.21). Hence, given a positive-semidefinite kernel $k$, we will use the notation $\langle \cdot, \cdot \rangle_k$ and $\|\cdot\|_k$ to denote the inner product and the norm, respectively, in the corresponding RKHS $\mathcal{H}_k$.

**Definition 2** (Strong convexity). *A differentiable function $f : \mathcal{H} \to \mathbb{R}$ on a Hilbert space $\mathcal{H}$ is $\alpha$-strongly convex over $\mathcal{S} \subseteq \mathcal{H}$, for a given $\alpha > 0$, if:*

$$\forall h, h' \in \mathcal{S}, \quad f(h) \geq f(h') + \langle \nabla f(h'), h - h'\rangle + \frac{\alpha}{2}\|h - h'\|_\mathcal{H}^2.$$

**Definition 3** (Smoothness). *A function $f : \mathcal{H} \to \mathcal{Y}$ between Hilbert spaces $\mathcal{H}$ and $\mathcal{Y}$ is $\eta$-smooth over $\mathcal{S} \subseteq \mathcal{H}$, for a given $\eta > 0$, if:*

$$\forall h, h' \in \mathcal{S}, \quad \|f(h) - f(h')\|_\mathcal{Y} \leq \eta\|h - h'\|_\mathcal{H}. \tag{28}$$

**Definition 4** (Sub-Gaussianity). *A random variable $\xi$ taking values in a Hilbert space $\mathcal{H}$ is said to be $\Sigma$-sub-Gaussian, given a positive-definite trace-class operator $\Sigma : \mathcal{H} \to \mathcal{H}$, if:*

$$\forall h \in \mathcal{H}, \quad \mathbb{E}[\exp\langle h, \xi\rangle] \leq \exp\left(\frac{1}{2}\langle h, \Sigma h\rangle\right). \tag{29}$$

*Likewise, a $\mathcal{H}$-valued stochastic process $\{\xi_n\}_{n=1}^{\infty}$ adapted to the filtration $\{\mathfrak{F}_n\}_{n=0}^{\infty}$ is conditionally $\Sigma$-sub-Gaussian if the following almost surely holds:*

$$\forall h \in \mathcal{H}, \quad \mathbb{E}[\exp\langle h, \xi_n\rangle \mid \mathfrak{F}_{n-1}] \leq \exp\left(\frac{1}{2}\langle h, \Sigma h\rangle\right), \quad \forall n \in \mathbb{N}. \tag{30}$$

## B.2 Auxiliary results

**Lemma 2.** *Let $g : \mathcal{X} \times \Theta \to \mathbb{R}$ represent a class of models parameterized by $\theta \in \Theta$. Assume that $g(\mathbf{x}; \cdot) \in \mathcal{H}_{\Theta}$, for all $\mathbf{x} \in \mathcal{X}$, where $\mathcal{H}_{\Theta}$ is a reproducing kernel Hilbert space associated with a positive-definite kernel $k_{\Theta} : \Theta \times \Theta \to \mathbb{R}$. It then follows that:*

$$\mathcal{H}_g := \{h : \mathcal{X} \to \mathbb{R} \mid \exists w \in \mathcal{H}_{\Theta} : h(\mathbf{x}) = \langle w, g(\mathbf{x}, \cdot)\rangle_{\mathcal{H}_{\Theta}}, \forall \mathbf{x} \in \mathcal{X}\} \tag{31}$$

*equipped with the norm:*

$$\|h\|_{\mathcal{H}_g} := \inf\{\|w\|_{\mathcal{H}_{\Theta}} : w \in \mathcal{H}_{\Theta}, h(\mathbf{x}) = \langle w, g(\mathbf{x}, \cdot)\rangle_{\mathcal{H}_{\Theta}}, \forall \mathbf{x} \in \mathcal{X}\} \tag{32}$$

*constitutes the unique RKHS where $k_g : (\mathbf{x}, \mathbf{x}') \mapsto \langle g(\mathbf{x}, \cdot), g(\mathbf{x}', \cdot)\rangle_{\mathcal{H}_{\Theta}}$ is the reproducing kernel.*

*Proof.* This is a direct application of classic RKHS results (e.g., Steinwart & Christmann, 2008, Thm. 4.21) where we are treating $\phi : \mathbf{x} \mapsto g(\mathbf{x}, \cdot)$ as a feature map mapping into an existing Hilbert space $\mathcal{H}_{\Theta}$ and taking advantage of its structure to define a new one. □

*Remark 2.* The RKHS $\mathcal{H}_g$ described above has the special property that, for any $\theta \in \Theta$, the RKHS norm of the model is given by:

$$\|g(\cdot, \theta)\|_{\mathcal{H}_g}^2 = k_{\Theta}(\theta, \theta), \tag{33}$$

since $\langle k_{\Theta}(\cdot, \theta), g(\mathbf{x}, \cdot)\rangle_{\mathcal{H}_{\Theta}} = g(\mathbf{x}, \theta)$ for all $\mathbf{x} \in \mathcal{X}$, and $k_{\Theta}(\cdot, \theta)$ is the unique representation of the evaluation functional at $\theta$ in the RKHS $\mathcal{H}_{\Theta}$. The rest follows from the definition in Equation 32. Hence, each choice of $k_{\Theta}$ gives us a potential RKHS norm regularizer.

**Lemma 3** (Abbasi-Yadkori, 2012, Cor. 3.6). *Let $\{\mathfrak{F}_t\}_{t=0}^{\infty}$ be an increasing filtration, $\{\epsilon_t\}_{t=1}^{\infty}$ be a real-valued stochastic process, and $\{\phi_t\}_{t=1}^{\infty}$ be a stochastic process taking values in a separable real Hilbert space $\mathcal{H}$, with both processes adapted to the filtration. Assume that $\{\phi_t\}_{t=1}^{\infty}$ is also predictable, i.e., $\phi_t$ is $\mathfrak{F}_{t-1}$-measurable, and that $\epsilon_t$ is conditionally $\sigma_\epsilon^2$-sub-Gaussian, for all $t \in \mathbb{N}$. Then, given any $\delta \in (0, 1)$, with probability at least $1 - \delta$,*

$$\forall t \in \mathbb{N}, \quad \left\|\sum_{i=1}^{t} \epsilon_i \phi_i\right\|_{(V + \Phi_t \Phi_t^{\mathsf{T}})^{-1}}^2 \leq 2\sigma_\epsilon^2 \log\left(\frac{\det(\mathbf{I} + \Phi_t^{\mathsf{T}} V^{-1} \Phi_t)^{\frac{1}{2}}}{\delta}\right),$$

*for any positive-definite operator $V \succ 0$ on $\mathcal{H}$, and where we set $\Phi_t := [\phi_1, \ldots, \phi_t]$.*

**Lemma 4** (GP variance upper bound (Steinberg et al., 2025, Lem. E.5)). *Let $\{\mathbf{x}_n\}_{n\geq 1}$ be a sequence of $\mathcal{X}$-valued random variables adapted to the filtration $\{\mathfrak{F}_n\}_{n\geq 1}$. For a given $\mathbf{x} \in \mathcal{X}$, assume that the following holds:*

$$\exists T_* \in \mathbb{N} : \quad \forall T \geq T_*, \quad \sum_{n=1}^{T} \mathbb{P}[\mathbf{x}_n = \mathbf{x} \mid \mathfrak{F}_{n-1}] \geq b_T > 0, \tag{34}$$

*for a some sequence of lower bounds $\{b_n\}_{n\in\mathbb{N}}$. Then, for a bounded kernel $k : \mathcal{X} \times \mathcal{X} \to \mathbb{R}$ given observations at $\{\mathbf{x}_i\}_{i=1}^{n}$, the following holds with probability 1:*

$$\sigma_n^2(\mathbf{x}) \in \mathcal{O}(b_n^{-1}). \tag{35}$$

*In addition, if $b_n \to \infty$, then $\lim_{n\to\infty} b_n \sigma_n^2(\mathbf{x}) \leq \sigma_\epsilon^2$.*

### B.3 MAIN ASSUMPTIONS

**Loss.** Given data $\mathcal{D}_n := \{\mathbf{x}_i, y_i\}_{i=1}^n$, for the analysis, we consider loss functionals defined over an RKHS $\mathcal{H}_k$ with the form:

$$L_n(g) = R_n(g) + \sum_{i=1}^n w_i \ell(m_i(g), z_i), \quad g \in \mathcal{H}_k, \tag{36}$$

where $\ell : \mathbb{R} \times \mathbb{R} \to \mathbb{R}$ is a fixed deterministic function, $w_i > 0$ represents importance sampling weights (or $w_i \propto 1$ when importance weights are not used), $m_i : \mathcal{H}_k \to \mathbb{R}$ represents a bounded linear observation functional (e.g., $m_i(g) = g(\mathbf{x}_i)$, or $m_i(g) = g(\mathbf{x}_{i,1}) - g(\mathbf{x}_{i,2})$), $z_i$ is given by utility evaluations (e.g., $z_i = u(y_i)$, or $z_{n,i} = u(y_{i,1}) - u(y_{i,2})$), for $i \in \{1, \ldots, n\}$, and $R_n : \mathcal{H}_k \to \mathbb{R}$ is a regularization functional.

Considering the loss structure above, the target loss functional is given by:

$$L_*(g) = \int \mathbb{E}[\ell(m(g), z)] \mu(\mathrm{d}m). \tag{37}$$

For the forward KL loss, for example, the target loss is simply $-\int_{\mathcal{X}} p_0(\mathbf{x}) a(\mathbf{x}) g(\mathbf{x}) \, \mathrm{d}\mu(\mathbf{x})$, where $\mu$ is a suitable base measure over $\mathcal{X}$ with respect to which the probability densities are defined (e.g., the counting measure if $\mathcal{X}$ is discrete). We then define constrained and unconstrained targets as:

$$g_* \in \underset{g \in \mathcal{H}_k : \int e^g \, \mathrm{d}\mu = 1}{\operatorname{argmin}} L_*(g) \tag{38}$$

$$\tilde{g}_* \in \underset{g \in \mathcal{H}_k}{\operatorname{argmin}} L_*(g). \tag{39}$$

**Assumption A1** (RKHS). *There exists a reproducing kernel Hilbert space $\mathcal{H}_k$, associated with a positive-definite kernel $k : \mathcal{X} \times \mathcal{X} \to \mathbb{R}$, which is bounded, $\sup_{\mathbf{x} \in \mathcal{X}} k(\mathbf{x}, \mathbf{x}) \leq b_k^2$ for a given $b_k > 0$, such that $g_*$, constants, and the models can be found as elements of $\mathcal{H}_k$, i.e., $\{g(\cdot; \theta) \mid \theta \in \Theta\} \subset \mathcal{H}_k$.*

The assumption above allows us to consider functions $g_*$ that cannot be perfectly approximated by the model, though which still lie in the same underlying Hilbert space $\mathcal{H}_k$. The reproducing kernel assumption is also mild, as it simply means that function evaluations are continuous (i.e., well behaved), which cannot usually be guaranteed in other types of Hilbert spaces, such as, e.g., $\mathcal{L}_2$-spaces. In fact, we can always find a RKHS that contains the set of models under mild assumptions, such as the minimal construction in Lemma 2. The inclusion of constant functions allows for the unnormalized target $\tilde{g}_*$ to also lie in $\mathcal{H}_k$.

*Remark* 3. If an RKHS containing the model class (e.g., $\mathcal{H}_g$ in Lemma 2) is too small to contain $g_*$, we can always combine two RKHS to produce a third one containing all the elements of the two. For instance, if $g_* \in \mathcal{H}_* \neq \mathcal{H}_g$ with kernel $k_* : \mathcal{X} \times \mathcal{X} \to \mathbb{R}$, we can define $k := k_* + k_g$, so that $\mathcal{H}_k := \mathcal{H}_* \oplus \mathcal{H}_g$ is also a RKHS (Steinwart & Christmann, 2008; Saitoh & Sawano, 2016). Such $\mathcal{H}_*$ can be minimal, as any function $g_*$ defines a kernel $k_*(\mathbf{x}, \mathbf{x}') = g_*(\mathbf{x}) g_*(\mathbf{x}')$ for the RKHS formed by the function's linear span $\mathcal{H}_* := \{\alpha g_* \mid \alpha \in \mathbb{R}\}$.

**Assumption A2** (Regularization). *The regularizer $R_n$ is predictable, $\lambda_n$-strongly convex, twice differentiable, and $\mathcal{O}(\lambda_n)$-smooth, for all $n \geq 1$, where $\{\lambda_n\}_{n=1}^\infty$ is a non-decreasing sequence of positive real numbers, such that $\lambda_n$ is at most poly-logarithmic with $n$.*

**Regularization.** Common choices of regularization scheme, such as the squared norm, suffice to satisfy Assumption A2. Strong convexity does not require a function to be twice differentiable, but such assumption greatly simplifies our analysis and is common in modern deep learning frameworks. Sublinearity of $\lambda_n$ allows for the effect of regularization to disappear as the dataset grows, so that the empirical $L_n$ can converge to the target $L_*$ as $n \to \infty$. Despite the definition of a regularization functional over the whole of $\mathcal{H}_k$, following Remark 2, we can use any positive-definite kernel $k_\Theta : \Theta \times \Theta \to \mathbb{R}$ compatible with Lemma 2 to set $R_n$ such that, over the model space:

$$R_n(g_\theta) = \frac{\lambda_n}{2} \|g_\theta\|_{\mathcal{H}_g}^2 = \frac{\lambda_n}{2} k_\Theta(\theta, \theta), \tag{40}$$

which allows for differentiation with respect to $\theta$, and whose Hessian $\nabla_g^2 R_n(g) = \lambda_n I$ shows that $R_n$ is strongly convex in $\mathcal{H}_k$ for $\lambda_n > 0$. In this case, a quadratic regularization penalty $\|\theta\|_2^2$

corresponds to the assumption of a linear kernel, i.e., $k_\Theta(\theta, \theta') = \theta \cdot \theta'$, which might appear quite restrictive, as it assumes that our models are linear functions of the parameters. However, note that, for overparameterized neural networks, at the infinite-width limit the model tends to show linearity in the parameters (Jacot et al., 2018). If we want to be more parsimonious, alternatively, we can choose $k_\Theta$ as a universal kernel, such as the squared exponential, yet preferably not translation invariant, so that $k_\Theta(\theta, \theta)$ is not simply a constant. One kernel satisfying such assumption would be the exponential dot-product kernel $k_\Theta(\theta, \theta') := \exp(\theta \cdot \theta')$, which is universal for continuous functions over compact subsets of $\Theta$. Nevertheless, we do not impose restrictions on the form of the regularization term $R_n$ other than Assumption A2. Lastly, $R_n$ can also be stochastic as:

$$R_n(g) = \frac{\lambda_n}{2} \|g - g_{n,0}\|_k^2, \tag{41}$$

where $\|\cdot\|_k$ denotes the norm in a general $\mathcal{H}_k$ satisfying Assumption A1, and $g_{n,0}$ may correspond to a random initialization or a data-dependent term (e.g., the previous set of optimal parameters).

**Assumption A3** (Loss). *For any $y \in \mathbb{R}$, the point loss $\ell_z := \ell(\cdot, y) : \mathbb{R} \to \mathbb{R}$ is $\alpha_\ell$-strongly convex, twice differentiable, and has $\eta_\ell$-smooth first-order derivatives. In addition, given any $m \in \mathcal{H}_k$, we assume the first-order derivative $\dot{\ell}_z(m(\tilde{g}_*))$ is conditionally $\sigma_\ell^2$-sub-Gaussian when $y \sim p(y|m)$.*

Regarding sub-Gaussianity, the pointwise derivative of the balanced forward KL loss is given by $\dot{\ell}_z(m(g)) = -u + e^{g(\mathbf{x})}$. Therefore, we have that:

$$\mathbb{E}[\dot{\ell}_z(m(g)) \mid \mathbf{x}] = e^{g(\mathbf{x})} - a(\mathbf{x}), \tag{42}$$

which yields $\tilde{g}_*(\mathbf{x}) = \log a(\mathbf{x})$ at the unconstrained minimizer $\tilde{g}_*$ under mild assumptions on the kernel and the acquisition function. Therefore, if observation noise is light tailed or the utilities are bounded, the pointwise derivatives can shown to be sub-Gaussian. The second derivative is simply $e^{g(\mathbf{x})}$, which is strictly positive, yielding an approximate strong convexity (when restricted to any bounded subset of the realizations of $g(\mathbf{x})$). The original Bradley-Terry model in the DPO paper (Rafailov et al., 2023) is not strongly convex, but its robust version (Chowdhury et al., 2024), which accounts for preference noise, can be shown to satisfy strong convexity and smoothness.

**Assumption A4** (Weights). *We have that $\inf_{n \in \mathbb{N}} w_n \geq b_w$ and $\sup_{n \in \mathbb{N}} w_n \leq \bar{b}_w$ almost surely, for constants $\bar{b}_w, b_w > 0$.*

**Assumption A5** (Density). *The model class is dense in the normalized subset of $\mathcal{H}_k$:*

$$\mathcal{C} := \left\{ g \in \mathcal{H}_k \;\middle|\; \int e^g \, \mathrm{d}\mu = 1 \right\}.$$

These last two assumptions allow us to bound the randomness of the loss gradients and ensure that the optimal models converge to the target distribution. Density here simply means that, for any $g \in \mathcal{C}$, we can find a sequence $\{\theta_i\}_{i=1}^\infty$ such that $g_{\theta_i} \to g$ as $i \to \infty$. Alternatively, the latter can be expressed as, for any $g \in \mathcal{C}$ and any given $\Delta > 0$, there exists $\theta_\Delta \in \Theta$ such that $\|g - g_{\theta_\Delta}\|_k \leq \Delta$. Hence, density is a universal approximation condition within the normalized subset of $\mathcal{H}_k$. As $\mathcal{H}_k$ is just a theoretical construct for us, and not fixed by the algorithm, so that we can even span $\mathcal{H}_k$ from the model class (see Lemma 2), Assumption A5 is relatively mild.

We can now analyze the approximation error with respect to $g_*$ for the following estimators:[1]

$$\theta_n \in \underset{\theta \in \Theta}{\arg\min} \, L_n(g_\theta) \tag{43}$$

$$g_n \in \underset{g \in \mathcal{H}_k}{\arg\min} \, L_n(g). \tag{44}$$

The first one gives us the best parametric approximation $g_{\theta_n}$ based on the data and is what our algorithm will use. The second estimator corresponds to the non-parametric approximation, which we will use as a tool for our analysis, and not assume as a component of the algorithm. The assumptions above allow us to bound distances between these estimators and the true $g_*$ as a function of the loss and gradient values. We also consider the normalized version of $g_n$ as:

$$\hat{g}_n(\mathbf{x}) := g_n(\mathbf{x}) - \log \int e^{g_n} \, \mathrm{d}\mu, \quad \mathbf{x} \in \mathcal{X}, \tag{45}$$

which we use as a reference point to bound distances to the parametric models and to the target $g_*$.

---

[1]We are implicitly assuming that such global optima exist. This is true for the optimization in $\mathcal{H}_k$, as $L_n$ is strongly convex over it, but that is not guaranteed for the optimization over the parameter space $\Theta$.

### B.4 MAIN RESULTS

**Lemma 1.** *Let assumptions A2 to A4 be satisfied. Then,*

$$\frac{1}{2}\|g - g_n\|_{H_n}^2 \le L_n(g) - L_n(g_n) \le \frac{1}{2}\|\nabla L_n(g)\|_{H_n^{-1}}^2,$$

*where $H_n : \mathcal{H}_k \to \mathcal{H}_k$ is an operator-valued lower bound on the Hessian of the loss $L_n$:*

$$\forall g \in \mathcal{H}_k, \quad \nabla^2 L_n(g) \succeq H_n := \lambda_n I + \alpha_\ell b_w \sum_{i=1}^n m_i \otimes m_i.$$

*Proof of Lemma 1.* The Hessian of the individual losses can be lower bounded by:

$$\forall g \in \mathcal{H}_k, \qquad \nabla_g^2 \ell(m(g), z) = \ddot{\ell}_z(m(g)) \nabla_g m(g) \otimes \nabla_g m(g) + \dot{\ell}_z(m(g)) \nabla_g^2 m(g)$$
$$= \ddot{\ell}_z(m(g)) m \otimes m \tag{46}$$
$$\succeq \alpha_\ell m \otimes m, \qquad \forall z \in \mathbb{R}, \ \forall m \in \mathcal{M},$$

given that $\nabla_g m(g) = \nabla_g \langle g, m \rangle_k = m$. Combining this result with Assumption A2 and A4, we get:

$$\forall g \in \mathcal{H}_k, \qquad \nabla_g^2 L_n(g) \succeq \lambda_n I + \alpha_\ell b_w \sum_{i=1}^n m_i \otimes m_i =: H_n. \tag{47}$$

Now applying a first order Taylor expansion to $L_n$ at any $g \in \mathcal{H}_k$, the error term is controlled by the Hessian $\nabla^2 L_n(\bar{g}_n)$ at an intermediate point $\bar{g}_n = sg_n + (1-s)g$, for some $s \in [0,1]$. Hence, expanding $L_n$ around $g_n$, we have that:

$$\forall g \in \mathcal{H}_k, \quad L_n(g) - L_n(g_n) = \langle \nabla L_n(g_n), g - g_n \rangle + \frac{1}{2}\|g - g_n\|_{\nabla^2 L_n(\bar{g}_n)}^2$$
$$\ge \frac{1}{2}\|g - g_n\|_{H_n}^2, \tag{48}$$

where we applied the Hessian inequality (47) and the fact that $\nabla L_n(g_n) = 0$, as $g_n$ is a minimizer. Thus, the lower bound in Lemma 1 follows. Conversely, expanding $L_n$ around any $g$ and evaluating at $g_n$, we have:

$$\forall g \in \mathcal{H}_k, \quad L_n(g_n) = L_n(g) + \langle \nabla L_n(g), g_n - g \rangle_k + \frac{1}{2}\|g_n - g\|_{\nabla^2 L_n(\bar{g}_n')}^2. \tag{49}$$

Rearranging terms yields:

$$\forall g \in \mathcal{H}_k, \quad L_n(g) - L_n(g_n) = \langle \nabla L_n(g), g - g_n \rangle_k - \frac{1}{2}\|g - g_n\|_{\nabla^2 L_n(\bar{g}_n')}^2$$
$$\le \sup_{\bar{g} \in \mathcal{H}_k} \langle \nabla L_n(g), \bar{g} \rangle_k - \frac{1}{2}\|\bar{g}\|_{\nabla^2 L_n(\bar{g}_n')}^2 \tag{50}$$
$$\le \sup_{\bar{g} \in \mathcal{H}_k} \langle \nabla L_n(g), \bar{g} \rangle_k - \frac{1}{2}\|\bar{g}\|_{H_n}^2,$$

whose right-hand side is strongly concave and has therefore a unique maximizer at:

$$\bar{g} = H_n^{-1} \nabla L_n(g). \tag{51}$$

Replacing this result into the previous equation finally leads us to the upper bound in Lemma 1. $\square$

**Lemma 5.** *Consider the setting in Lemma 1. Then,*

$$\forall n \in \mathbb{N}, \quad |\langle m, \tilde{g}_* \rangle_k - \langle m, g_n \rangle_k| \le \|m\|_{H_n^{-1}} \beta_n(\delta), \ \forall m \in \mathcal{H}_k,$$

*where $\beta_n(\delta) := \lambda_n^{-1/2} \|\nabla R_n(\tilde{g}_*)\|_k + \sigma_\ell \bar{b}_w \sqrt{2(b_w \alpha_\ell)^{-1} \log(\det(\mathbf{I} + b_w \alpha_\ell \lambda_n^{-1} M_n^{\mathsf{T}} M_n)^{1/2}/\delta)}$, and $M_n := [m_1, \dots, m_n]$.*

*Proof.* By Lemma 1, for any $g \in \mathcal{H}_k$, we have that:

$$
\begin{aligned}
|\langle m, g_n \rangle_k - \langle m, g \rangle_k| &= |\langle m, g_n - g \rangle_k| \\
&= |\langle H_n^{-1/2} m, H_n^{1/2}(g_n - g) \rangle_k| \\
&\leq \|H_n^{-1/2} m\| \|H_n^{1/2}(g_n - g)\| \\
&= \|m\|_{H_n^{-1}} \|g_n - g\|_{H_n} \\
&\leq \|m\|_{H_n^{-1}} \|\nabla L_n(g)\|_{H_n^{-1}}, \quad \forall m \in \mathcal{H}_k,
\end{aligned}
\tag{52}
$$

where the first inequality follows by Cauchy-Schwarz, and the last is due to Lemma 1. Expanding the gradient term, we have:

$$
\begin{aligned}
\|\nabla L_n(g)\|_{H_n^{-1}} &= \left\| \nabla R_n(g) + \sum_{i=1}^{n} w_i \dot{\ell}_{y_i}(\langle m_i, g \rangle_k) m_i \right\|_{H_n^{-1}} \\
&\leq \|\nabla R_n(g)\|_{H_n^{-1}} + \left\| \sum_{i=1}^{n} w_i \dot{\ell}_{y_i}(\langle m_i, g \rangle_k) m_i \right\|_{H_n^{-1}} \\
&\leq \frac{1}{\sqrt{\lambda}} \|\nabla R_n(g)\|_k + \left\| \sum_{i=1}^{n} w_i \dot{\ell}_{y_i}(\langle m_i, g \rangle_k) m_i \right\|_{H_n^{-1}},
\end{aligned}
\tag{53}
$$

where we applied the triangle inequality to obtain the second line and the fact that $H_n \succ \lambda_n I$ implies $H_n^{-1} \prec \lambda_n^{-1} I$ led to the last line. For $g := \tilde{g}_*$, we can then apply Lemma 3 to the noisy sum above by setting $\mathfrak{F}_t$ as the $\sigma$-algebra generated by the random variables $\{m_i, y_i\}_{i=1}^{t}$ and $m_{t+1}$. Then $\epsilon_t := \frac{1}{\sqrt{b_w \alpha_\ell}} w_t \dot{\ell}_{y_t}(\langle \tilde{g}_*, m_t \rangle_k)$ is conditionally $\sigma_\epsilon^2$-sub-Gaussian by Assumption A3 with $\sigma_\epsilon^2 := \frac{\bar{b}_w^2 \sigma_\ell^2}{b_w \alpha_\ell}$, since $w_t \leq \bar{b}_w$ by Assumption A4, and $\phi_t := \sqrt{b_w \alpha_\ell} m_t$ is predictable, for all $t \in \mathbb{N}$. An application of Lemma 3 finally leads us to:

$$
\left\| \sum_{i=1}^{n} \dot{\ell}_{y_i}(\langle m_i, \tilde{g}_* \rangle_k) m_i \right\|_{H_n^{-1}}^2 \leq \frac{2\bar{b}_w^2 \sigma_\ell^2}{b_w \alpha_\ell} \log\left( \frac{\det(\mathbf{I} + b_w \alpha_\ell \lambda_n^{-1} M_n^\mathsf{T} M_n)^{\frac{1}{2}}}{\delta} \right)
\tag{54}
$$

which holds uniformly over all $n \in \mathbb{N}$ with probability at least $1 - \delta$. Hence, it follows that:

$$
\forall n \in \mathbb{N}, \quad \|\nabla L_n(\tilde{g}_*)\|_{H_n^{-1}} \leq \frac{1}{\sqrt{\lambda}} \|\nabla R_n(\tilde{g}_*)\|_k + \left\| \sum_{i=1}^{n} w_i \dot{\ell}_{y_i}(\langle m_i, \tilde{g}_* \rangle_k) m_i \right\|_{H_n^{-1}} \leq \beta_n(\delta), \tag{55}
$$

with probability at least $1 - \delta$. Replacing this result into Equation 52 yields the main result. $\square$

**Theorem 1.** *Let assumptions A1 to A5 hold. Then, given any $\delta \in (0, 1)$,*

$$
\forall n \in \mathbb{N}, \quad |\langle m, g_* \rangle_k - \langle m, g_{\theta_n} \rangle_k| \leq \beta_n(\delta)(\|m\|_{H_n^{-1}} + 2|\langle m, \iota \rangle_k| v_n(\bar{q}_n)), \quad \forall m \in \mathcal{H}_k,
$$

*which holds with probability at least $1 - \delta$, where $\beta_n(\delta)$ is given by Lemma 5, where $\iota(\mathbf{x}) = 1$ denotes the unit constant function, $v_n(q) := \mathbb{E}_{\mathbf{x} \sim q}\left[ \|\phi(\mathbf{x})\|_{H_n^{-1}} \right]$, and $\bar{q}_n := \frac{p_u^* + q_n}{2}$.*

*Proof of Theorem 1.* Fix any $m \in \mathcal{H}_k$ and $n \in \mathbb{N}$, the approximation error can then be decomposed as:

$$
|\langle m, g_* \rangle_k - \langle m, g_{\theta_n} \rangle_k| \leq |\langle m, g_* - \hat{g}_n \rangle_k| + |\langle m, g_{\theta_n} - \hat{g}_n \rangle_k|.
\tag{56}
$$

Let $\iota \in \mathcal{H}_k$ denote the unit constant function, i.e., $\iota(\mathbf{x}) = 1$, for all $\mathbf{x} \in \mathcal{X}$. The difference between normalized functions is then such that:

$$
\begin{aligned}
|\langle m, g_* - \hat{g}_n \rangle_k| &= \left| \left\langle m, \tilde{g}_* - g_n + \iota \left( \log \int e^{g_n} \, \mathrm{d}\mu - \log \int e^{\tilde{g}_*} \, \mathrm{d}\mu \right) \right\rangle_k \right| \\
&\leq |\langle m, \tilde{g}_* - g_n \rangle_k| + |\langle m, \iota \rangle_k| \left| \log \int e^{g_n} \, \mathrm{d}\mu - \log \int e^{\tilde{g}_*} \, \mathrm{d}\mu \right|
\end{aligned}
\tag{57}
$$

We can apply Jensen's inequality to show that the second term on the right-hand side of the inequality is bounded by the expected pointwise error between $g_n$ and $\tilde{g}_*$. To reduce notation clutter, let $q_* := p_u^* = \exp g_*$ and $\mathbb{E}_q[h] := \mathbb{E}_{\mathbf{x} \sim q}[h(\mathbf{x})]$. Indeed, therefore, we have that:

$$
\begin{aligned}
\mathbb{E}_{q_*}[\tilde{g}_* - g_n] &= \mathbb{E}_{q_*}\left[-\log\left(\frac{e^{g_n}}{e^{\tilde{g}_*}}\right)\right] \\
&\geq -\log \mathbb{E}_{q_*}\left[\frac{e^{g_n}}{e^{\tilde{g}_*}}\right] \\
&= -\log\left(\frac{1}{\int e^{\tilde{g}_*}\,\mathrm{d}\mu}\int e^{\tilde{g}_*}\frac{e^{g_n}}{e^{\tilde{g}_*}}\,\mathrm{d}\mu\right) \\
&= \log\int e^{\tilde{g}_*}\,\mathrm{d}\mu - \log\int e^{g_n}\,\mathrm{d}\mu
\end{aligned}
\tag{58}
$$

by Jensen's inequality on the convex $-\log(\cdot)$. Similarly, repeating the same steps for $\mathbb{E}_{\hat{q}_n}[g_n - \tilde{g}_*]$, where $\hat{q}_n(\mathbf{x}) = e^{\hat{g}_n(\mathbf{x})} = \frac{e^{g_n(\mathbf{x})}}{\int e^{g_n}\,\mathrm{d}\mu}$, we get:

$$
\mathbb{E}_{\hat{q}_n}[g_n - \tilde{g}_*] \geq \log\int e^{g_n}\,\mathrm{d}\mu - \log\int e^{\tilde{g}_*}\,\mathrm{d}\mu.
\tag{59}
$$

Combining the two inequalities yields:

$$
\begin{aligned}
\left|\log\int e^{g_n}\,\mathrm{d}\mu - \log\int e^{\tilde{g}_*}\,\mathrm{d}\mu\right| &\leq \max\{\mathbb{E}_{q_*}[\tilde{g}_* - g_n], \mathbb{E}_{\hat{q}_n}[g_n - \tilde{g}_*]\} \\
&\leq |\mathbb{E}_{q_*}[\tilde{g}_* - g_n]| + |\mathbb{E}_{\hat{q}_n}[g_n - \tilde{g}_*]|.
\end{aligned}
\tag{60}
$$

We can now bound the expectations as:

$$
\begin{aligned}
\forall q \in \mathcal{P}(\mathcal{X}), \quad |\mathbb{E}_q[\tilde{g}_* - g_n]| &= |\mathbb{E}_{\mathbf{x}\sim q}[\tilde{g}_*(\mathbf{x}) - g_n(\mathbf{x})]| \\
&= |\mathbb{E}_{\mathbf{x}\sim q}[\langle\tilde{g}_* - g_n, \phi(\mathbf{x})\rangle_k]| \\
&= |\langle\tilde{g}_* - g_n, \mathbb{E}_{\mathbf{x}\sim q}[\phi(\mathbf{x})]\rangle_k| \\
&\leq \beta_n(\delta)\|\mathbb{E}_q[\phi(\mathbf{x})]\|_{H_n^{-1}} \\
&\leq \beta_n(\delta)\mathbb{E}_q\left[\|\phi(\mathbf{x})\|_{H_n^{-1}}\right].
\end{aligned}
\tag{61}
$$

where the first inequality follows by Lemma 5 and the second is due to Jensen's. Therefore,

$$
\left|\log\int e^{g_n}\,\mathrm{d}\mu - \log\int e^{\tilde{g}_*}\,\mathrm{d}\mu\right| \leq \beta_n(\delta)\left(\mathbb{E}_{q_*}\left[\|\phi(\mathbf{x})\|_{H_n^{-1}}\right] + \mathbb{E}_{\hat{q}_n}\left[\|\phi(\mathbf{x})\|_{H_n^{-1}}\right]\right).
\tag{62}
$$

Finally, applying the bound above to Equation 57 followed by another application of Lemma 5, we obtain:

$$
|\langle m, g_* - \hat{g}_n\rangle_k| \leq \beta_n(\delta)\left(\|m\|_{H_n^{-1}} + |\langle m, \iota\rangle_k|\left(\mathbb{E}_{q_*}\left[\|\phi(\mathbf{x})\|_{H_n^{-1}}\right] + \mathbb{E}_{\hat{q}_n}\left[\|\phi(\mathbf{x})\|_{H_n^{-1}}\right]\right)\right).
\tag{63}
$$

For the remaining term in Equation 56, we have that:

$$
\begin{aligned}
|\langle m, g_{\theta_n}\rangle_k - \langle m, \hat{g}_n\rangle_k| &\leq \|m\|_{H_n^{-1}}\|g_{\theta_n} - \hat{g}_n\|_{H_n} \\
&\leq \|m\|_{H_n^{-1}}\sqrt{2(L_n(g_{\theta_n}) - L_n(\hat{g}_n))},
\end{aligned}
\tag{64}
$$

which follows from Lemma 1. From Assumption A5, we have that:

$$
\forall \Delta > 0, \quad \exists \theta_\Delta \in \Theta : \qquad \|g_{\theta_\Delta} - \hat{g}_n\|_k \leq \Delta.
\tag{65}
$$

At the optimum, we know that $L_n(g_{\theta_n}) \leq L_n(g_\theta)$, for all $\theta \in \Theta$. Therefore, as $\Delta \to 0$, by continuity, we have that:

$$
L_n(g_{\theta_n}) - L_n(\hat{g}_n) \leq L_n(g_{\theta_\Delta}) - L_n(\hat{g}_n) \to 0,
\tag{66}
$$

which implies $|\langle m, g_{\theta_n}\rangle_k - \langle m, \hat{g}_n\rangle_k| \to 0$ in Equation 64. Consequently, the density of the model class allows us to replace pointwise evaluations $\hat{g}_n(\mathbf{x}) = \langle\hat{g}_n, \phi(\mathbf{x})\rangle_k$ by $g_{\theta_n}(\mathbf{x})$, for $\mathbf{x} \in \mathcal{X}$, so that we can swap $\hat{q}_n$ in Equation 63 for $q_n(\mathbf{x}) = \exp g_{\theta_n}(\mathbf{x})$. The main result then follows. $\qquad\square$

Despite the model being potentially non-linear and the loss not being required to be least-squares, Theorem 1 shows that we recover the same kind of RKHS-based error bound found in the kernelized bandits literature (Chowdhury & Gopalan, 2017; Durand et al., 2018; Oliveira et al., 2021). Regarding the asymptotic rates, we make the following observations.

*Remark* 4. For a finite domain, $|\mathcal{X}| < \infty$, the RKHS becomes finite dimensional with $\dim(\mathcal{H}_k) = |\mathcal{X}|$, resembling a linear model. In this case, Srinivas et al. (2010) provides bounds for the Gram matrix log-determinant in $\beta_n(\delta)$ as $\mathcal{O}(|\mathcal{X}| \log n)$, yielding:

$$\beta_n(\delta) \in \mathcal{O}(\sqrt{|\mathcal{X}| \log n}).$$

**Predictive variance.** We can show a connection between $\|m\|_{H_n^{-1}}$ and a GP predictive variance. By an application of Woodbury's identity, letting $\widehat{\alpha_\ell} := \alpha_\ell b_w$, we have that:

$$
\begin{aligned}
\|m\|_{H_n^{-1}}^2 &= m^\mathsf{T}(\lambda_n I + \widehat{\alpha_\ell} M_n M_n^\mathsf{T})^{-1} m \\
&= m^\mathsf{T}(\lambda_n^{-1} I - \lambda_n^{-2} M_n (\widehat{\alpha_\ell}^{-1}\mathbf{I} + \lambda_n^{-1} M_n^\mathsf{T} M_n)^{-1} M_n^\mathsf{T}) m \\
&= \lambda_n^{-1} m^\mathsf{T}(I - M_n(\lambda_n \widehat{\alpha_\ell}^{-1}\mathbf{I} + M_n^\mathsf{T} M_n)^{-1} M_n^\mathsf{T}) m \\
&= \lambda_n^{-1}(\|m\|_k^2 - m^\mathsf{T} M_n(\lambda_n \widehat{\alpha_\ell}^{-1}\mathbf{I} + M_n^\mathsf{T} M_n)^{-1} M_n^\mathsf{T} m),
\end{aligned}
\tag{67}
$$

If observations correspond to pointwise evaluations $m := k(\cdot, \mathbf{x}) = \phi(\mathbf{x})$ and $m_i := k(\cdot, \mathbf{x}_i) = \phi(\mathbf{x}_i)$, for $\mathbf{x} \in \mathcal{X}$ and $\{\mathbf{x}_i\}_{i=1}^n \subset \mathcal{X}$, we end up with:

$$
\begin{aligned}
\|m\|_{H_n^{-1}}^2 &= \|\phi(\mathbf{x})\|_{H_n^{-1}}^2 \\
&= \lambda_n^{-1}(k(\mathbf{x}, \mathbf{x}) - \mathbf{k}_n(\mathbf{x})^\mathsf{T}(\lambda_n \widehat{\alpha_\ell}^{-1}\mathbf{I} + \mathbf{K}_n)^{-1} \mathbf{k}_n(\mathbf{x})) \\
&= \lambda_n^{-1} \sigma_n^2(\mathbf{x}),
\end{aligned}
\tag{68}
$$

which corresponds to a scaled version of the posterior predictive variance $\sigma_n^2(\mathbf{x})$ of a GP (25). Moreover, the $\langle m, \iota \rangle_k$ term in Theorem 1 is simply $\langle \phi(\mathbf{x}), \iota \rangle_k = \iota(\mathbf{x}) = 1$, for $\mathbf{x} \in \mathcal{X}$, and the remaining terms depend on expectations of $\|\phi\|_{H_n^{-1}}$, which is also a function of the predictive variance $\sigma_n^2(\mathbf{x})$. We can then apply the auxiliary result from VSD (Lemma 4) to show that $\sigma_n^2(\mathbf{x})$ is $\mathcal{O}(n^{-1})$ asymptotically, whenever $q_{\theta_n}(\mathbf{x})$ has a positive lower bound, allowing for asymptotic convergence of the proposal distributions to the target $p_u^* = \exp g_*$. As a result, considering Remark 4, we get:

$$|g_*(\mathbf{x}) - g_n(\mathbf{x})| \in \mathcal{O}\left(\sqrt{\frac{|\mathcal{X}| \log n}{n}}\right),\tag{69}$$

which vanishes as $n \to \infty$. Therefore, the model converges to the target distribution $p_u^*$ under these assumptions. Nevertheless, the algorithm's regret still depends on how fast the time-dependent acquisition function $a_t$ concentrates around the optimum $\mathbf{x}^*$ and how much probability mass $q_t$ places away from $\mathbf{x}^*$ per BO round $t$, analyses which we leave as subject of future work.

### B.5 LOSS FUNCTIONS

In the following, we present further details on the derivation of the loss functions. We consider a fixed non-negative utility function $u : \mathbb{R} \to [0, \infty)$ and the corresponding acquisition function $a : \mathcal{X} \to \mathbb{R}$ defined as $a(\mathbf{x}) = \mathbb{E}[u(y)|\mathbf{x}]$, where the expectation is over the observations distribution.

**Forward KL.** Expanding the definition, we have:

$$
\begin{aligned}
\mathbb{D}_{\mathrm{KL}}(p_u^* || q) &= \mathbb{E}_{\mathbf{x} \sim p_u^*}[\log p_u^*(\mathbf{x}) - \log q(\mathbf{x})] \\
&= \mathbb{E}_{\mathbf{x} \sim p_u^*}[\log p_u^*(\mathbf{x})] - \mathbb{E}_{\mathbf{x} \sim p_u^*}[\log q(\mathbf{x})] \\
&= -\mathbb{H}[p_u^*] - \mathbb{E}_{\mathbf{x} \sim p_u^*}[\log q(\mathbf{x})].
\end{aligned}
\tag{70}
$$

Note that the entropy $\mathbb{H}[p_u^*]$ of the target is constant. Using importance sampling with a proposal $q_0$, the remaining expectation can be approximated as:

$$\mathbb{E}_{\mathbf{x} \sim p_u^*}[\log q(\mathbf{x})] = \mathbb{E}_{\mathbf{x} \sim q_0}\left[\frac{p_u^*(\mathbf{x})}{q_0(\mathbf{x})} \log q(\mathbf{x})\right] \approx \frac{1}{B}\sum_{i=1}^B \frac{p_u^*(\mathbf{x}_i)}{q_0(\mathbf{x}_i)} \log q(\mathbf{x}_i),\tag{71}$$

for a batch of $B \geq 1$ samples $\mathbf{x}_i \sim q_0$ sampled i.i.d. from the proposal $q_0$. Having $t \geq 1$ proposals, instead, we get:

$$\mathbb{E}_{\mathbf{x} \sim p_u^*}[\log q(\mathbf{x})] \approx \frac{1}{tB} \sum_{j=1}^{t} \sum_{i=1}^{B} \frac{p_u^*(\mathbf{x}_{j,i})}{q_{j-1}(\mathbf{x}_{j,i})} \log q(\mathbf{x}_{j,i}), \tag{72}$$

where $\{\mathbf{x}_{j,i}\}_{i=1}^{B} \overset{i.i.d.}{\sim} q_{j-1}$, for $j \in \{1, \ldots, t\}$. We do not have access to the full $p_u^*(\mathbf{x}) = \frac{a(\mathbf{x})p_0(\mathbf{x})}{\int_{\mathcal{X}} a(\mathbf{x}')p_0(\mathbf{x}') \, d\mu(\mathbf{x}')}$ due to the intractability of the normalization factor[2] $\int_{\mathcal{X}} a(\mathbf{x}')p_0(\mathbf{x}') \, d\mu(\mathbf{x}')$ nor the full acquisition function $a(\mathbf{x}) = \mathbb{E}[u(y)|\mathbf{x}]$, as we only observe noisy utilities $u(y_i)$. The normalization factor, however, is constant, and the acquisition function can be unbiasedly approximated via Monte Carlo. Using single-sample estimates for the latter, we then obtain our final form:

$$\mathbb{E}_{\mathbf{x} \sim p_u^*}[\log q(\mathbf{x})] \approx \frac{1}{tB} \sum_{j=1}^{t} \sum_{i=1}^{B} \frac{p_0(\mathbf{x}_{j,i})u(y_{j,i})}{q_{j-1}(\mathbf{x}_{j,i})} \log q(\mathbf{x}_{j,i}) = \frac{1}{n_t} \sum_{i=1}^{n_t} \frac{p_0(\mathbf{x}_i)u(y_i)}{q_{i-1}(\mathbf{x}_i)} \log q(\mathbf{x}_i), \tag{73}$$

where the latter follows by simple re-indexing, with $n_t := tB$. We can also drop the constant $\frac{1}{n_t}$ during optimization.

**Balanced forward KL.** The balanced forward KL arises from the definition of Bregman divergences:

$$\mathbb{D}(p||q) = \Psi(p) - \Psi(q) - \langle \nabla \Psi(q), p - q \rangle, \qquad p, q \in \Omega, \tag{74}$$

where $\Psi : \Omega \to \mathbb{R}$ is a convex function over a convex subset $\Omega$ of a vector space. In our case, $\Omega$ is the space $\mathcal{P}(\mathcal{X})$ of probability measures over the domain $\mathcal{X}$, which can be embedded as convex subset of $\mathcal{L}_2(\mu)$ when restricted to measures that have a square-integrable density with respect to the base measure $\mu$. Now consider the negative entropy functional:

$$\Psi(p) = \int_{\mathcal{X}} p(\mathbf{x}) \log p(\mathbf{x}) \, d\mu(\mathbf{x}). \tag{75}$$

Its functional gradient is given by:

$$\nabla \Psi(p) = 1 + \log p. \tag{76}$$

Thus, the Bregman divergence with this functional is:

$$\begin{aligned}
\mathbb{D}(p||q) &= \Psi(p) - \Psi(q) - \langle \nabla \Psi(q), p - q \rangle \\
&= \int_{\mathcal{X}} p(\mathbf{x}) \log p(\mathbf{x}) \, d\mu(\mathbf{x}) - \int_{\mathcal{X}} q(\mathbf{x}) \log q(\mathbf{x}) \, d\mu(\mathbf{x}) - \langle 1 + \log q, p - q \rangle_{\mathcal{L}_2(\mu)} \\
&= \int p \log p \, d\mu - \int q \log q \, d\mu - \int p - q \, d\mu - \int (p - q) \log q \, d\mu \\
&= \int p(\log p - \log q) \, d\mu - \int p \, d\mu + \int q \, d\mu.
\end{aligned} \tag{77}$$

We could cancel the constants $\int p \, d\mu = \int q \, d\mu = 1$. Instead, we keep the term $\int q \, d\mu$, which after an importance sampling approximation, allows us to have a non-banishing term for observations where $u(y) = 0$. Namely, using a proposal $q_0$, we have that:

$$\begin{aligned}
\mathbb{D}(p_u^*||q) &= \int q_0 \frac{p_u^*}{q_0}(\log p_u^* - \log q) \, d\mu + \int q_0 \frac{q}{q_0} \, d\mu - \int p_u^* \, d\mu \\
&= -\int q_0 \frac{p_u^*}{q_0} \log q \, d\mu + \int q_0 \frac{q}{q_0} \, d\mu + c,
\end{aligned} \tag{78}$$

where $c := \int q_0 \frac{p_u^*}{q_0} \log p_u^* \, d\mu - \int p_u^* \, d\mu$ is constant. Dropping constants and following a similar approach to the derivation of the forward KL loss, we get:

$$-\int p_u^* \log q \, d\mu + \int q \, d\mu \approx -\sum_{i=1}^{n_t} \frac{p_0(\mathbf{x}_i)u(y_i)}{q_{i-1}(\mathbf{x}_i)} \log q(\mathbf{x}_i) - \frac{q(\mathbf{x}_i)}{q_{i-1}(\mathbf{x}_i)}, \qquad t \geq 1. \tag{79}$$

---

[2]Recall that $\mu$ represents a base measure of the domain with respect to which the proposal densities are defined, i.e., the counting measure for a discrete domain (our experiments) or the Lebesgue measure for a continuous domain.

## C   ADDITIONAL EXPERIMENTAL DETAIL

This section presents algorithmic settings and implementation details for our experiments. We also present a summary of our experimental results in Section C.4 and ablation studies in Section C.5. In addition to the descriptions in this section, code for our experiments is available online at:

https://github.com/csiro-funml/generativebo

### C.1   TEXT OPTIMIZATION

We use the same annealing threshold scheme for setting $\tau_t$ as Steinberg et al. (2025, Eqn. 20), where we set $\eta$ such that we begin at $p_0 = 0.5$ we end at $p_T = 0.99$. For the proposal distribution, we found these short sequences best generated by the simple mean-field categorical model,

$$q(\mathbf{x}|\phi) = \prod_{m=1}^{M} \text{Categ}(x_m|\text{softmax}(\phi_m)) \tag{80}$$

where $x_m \in \mathcal{V}$ and $\phi_m \in \mathbb{R}^{|\mathcal{V}|}$, and we directly optimize $\phi$. VSD and CbAS use the simple MLP classifier guide in Figure 4.

```
Sequential(
    Embedding(
        num_embeddings=A,
        embedding_dim=16
    ),
    Dropout(p=0.1),
    Flatten(),
    LeakyReLU(),
    Linear(
        in_features=16 * M,
        out_features=64
    ),
    LeakyReLU(),
    Linear(
        in_features=64,
        out_features=1
    ),
)
```

```
Sequential(
    Embedding(
        num_embeddings=A,
        embedding_dim=E
    ),
    Dropout(p=0.2),
    Conv1d(
        in_channels=E,
        out_channels=C,
        kernel_size=Kc,
    ),
    LeakyReLU(),
    MaxPool1d(
        kernel_size=Kx,
        stride=Sx,
    ),
    Conv1d(
        in_channels=C,
        out_channels=C,
        kernel_size=Kc,
    ),
    LeakyReLU(),
    MaxPool1d(
        kernel_size=Kx,
        stride=Sx,
    ),
    Flatten(),
    LazyLinear(
        out_features=H
    ),
    LeakyReLU(),
    Linear(
        in_features=H,
        out_features=1
    ),
)
```

(a) MLP architecture                    (b) CNN architecture

Figure 4: Classifier architectures used for VSD and CbAS in the experiments using PyTorch syntax. A $= |\mathcal{V}|$, M $= M$, and we give all other parameters in Table 2 if not directly indicated.

## C.2 PROTEIN DESIGN

We use the same threshold function and setting for all of the protein design experiments as in Section C.1. However, these tasks require a more sophisticated generative model that can capture local and global relationships that relate to a protein's 3D structure. For this we use the auto-regressive (causal) transformer architecture also used in Steinberg et al. (2025),

$$q(\mathbf{x}|\phi) = \text{Categ}(x_1|\text{softmax}(\phi_1)) \prod_{m=2}^{M} q(x_m|x_{1:m-1}, \phi_{1:m}), \quad \text{where}$$

$$q(x_m|x_{1:m-1}, \phi_{1:m}) = \text{Categ}(x_m|\text{DTransformer}(x_{1:m-1}, \phi_{1:m})). \tag{81}$$

For the latter, see Algorithm 10 and 14 in Phuong & Hutter (2022) for maximum likelihood training and sampling implementation details, respectively. We give the architectural configuration for the transformers in each task in Table 1, and the classifier CNN used by VSD and CbAS is in Figure 4.

| Configuration | Stability | SASA | Ehrlich 15 | Ehrlich 32 | Ehrlich 64 |
|---|---|---|---|---|---|
| Layers | 2 | 2 | 2 | 2 | 2 |
| Feedforward network | 256 | 256 | 32 | 64 | 128 |
| Attention heads | 4 | 4 | 1 | 2 | 3 |
| Embedding size | 64 | 64 | 10 | 20 | 30 |

Table 1: Transformer backbone configuration.

| Configuration | Stability | SASA | Ehrlich 15 | Ehrlich 32 | Ehrlich 64 |
|---|---|---|---|---|---|
| E | 16 | 16 | 10 | 10 | 10 |
| C | 96 | 96 | 16 | 16 | 16 |
| Kc | 7 | 7 | 4 | 7 | 7 |
| Kx | 5 | 5 | 2 | 2 | 2 |
| Sx | 4 | 4 | 2 | 2 | 2 |
| H | 192 | 192 | 128 | 128 | 128 |

Table 2: CNN guide configuration for VSD and CbAS

We use the following Ehrlich function configurations:

$M = 15$: motif length = 4, no. motifs = 2, quantization = 4

$M = 32$: motif length = 4, no. motifs = 2, quantization = 4

$M = 64$: motif length = 4, no. motifs = 8, quantization = 4

## C.3 GENBO SETTINGS

Table 3 presents our settings for the different GenBO variants across experiments. The settings for our proposal models followed VSD's configurations. Our regularization scheme penalized the Euclidean distance between the model's parameters and their random initialization (He et al., 2015) with $R_n(\theta) := \lambda_n \|\theta - \theta_0\|_2^2$, using an annealed regularization factor $\lambda_n := \lambda_0 \log^2 n$, similar to Dai et al. (2022), which ensures enough exploration, while still $\frac{\lambda_n}{n} \to 0$ as $n \to \infty$, allowing for convergence to the optimal $\theta_*$. For threshold-based utilities, we mainly set the quantile threshold $\tau_t$ to follow an annealing schedule ranging from the 50th (i.e., the median) to the 99th percentile of the observations marginal distribution for both GenBO and VSD, where the percentile $\gamma_t$ corresponding the quantile is updated as $\gamma_t := \gamma_{t-1}^\eta$, where $\eta := \left(\frac{\log \gamma_T}{\log \gamma_0}\right)^{\frac{1}{T-1}} \in (0, 1)$.

## C.4 RESULTS SUMMARY

Besides the plots in section 6, we summarize the final results in Table 4 and 5.

| Acronym | Meaning |
|---|---|
| EI | Expected Improvement |
| PI | Probability of Improvement |
| sEI | Soft Expected Improvement, i.e., $\mathrm{softplus}(y - \tau)$ |
| fKL | Forward KL loss |
| bfKL | Balanced forward KL loss |
| rPL | Robust preference loss |
| MF | Mean-field categorical proposal model |
| Tfm | Transformer proposal model |
| fr | More frequent regularization (change in $\lambda_n$ schedule rate) |
| r0p10 | Base regularization factor set to $\lambda_0 := 0.1$ |
| exp | Exponential regularizer, i.e., $R_n(\theta) := \lambda_n \exp\|\theta - \theta_0\|_2^2$ |
| np | No (informative) prior, i.e., $p_0(\mathbf{x}) \propto 1$ |
| p | Pre-trained prior, learned from initial (randomly initialized) data $\mathcal{D}_0$ |
| lg | Importance weights |
| lr0p10 | Learning rate setting for training the generative model (e.g., 0.1 in this case) |
| pcmin0p50 | Minimum percentile for threshold $\tau_t$ annealing schedule (e.g., 50% in this case) |
| pcmax0p90 | Maximum percentile for threshold $\tau_t$ annealing schedule (e.g., 90% in this case) |

Table 3: GenBO experiment settings acronyms

|  | ALOHA | Ehrlich-15 | Ehrlich-32 | Ehrlich-64 |
|---|---|---|---|---|
| Random mut. | $3.80 \pm 0.40$ | | | |
| LaMBO-2 | | $0.19 \pm 0.17$ | $0.36 \pm 0.15$ | $0.95 \pm 0.02$ |
| CbAS | $2.20 \pm 0.40$ | $0.57 \pm 0.12$ | $0.61 \pm 0.10$ | $0.98 \pm 0.01$ |
| GA | | $0.45 \pm 0.12$ | $0.61 \pm 0.10$ | $0.98 \pm 0.01$ |
| VSD | $\mathbf{0.00 \pm 0.00}$ | $0.19 \pm 0.17$ | $0.32 \pm 0.09$ | $0.97 \pm 0.00$ |
| GenBO | $0.20 \pm 0.40$ | $\mathbf{0.00 \pm 0.00}$ | $\mathbf{0.28 \pm 0.16}$ | $\mathbf{0.94 \pm 0.02}$ |

Table 4: Final average regret (lower is better) for the best-performing variant of each method across the ALOHA (text optimization) and Ehrlich benchmarks

## C.5    ABLATIONS

We performed ablation studies on the annealing scheme and the evaluations batch size. We vary the minimum and maximum percentile for the threshold annealing settings of both GenBO (with PI utility) and VSD on the text optimization problem in Figure 5. Plots reveal a preference towards a more exploitative behavior for this simpler optimization problem. In Figure 6, we vary the evaluations batch size $B$ for GenBO on the Ehrlich benchmark problem of sequence length 32. As expected, larger evaluation batches lead to lower regret, though with higher variability across the candidates.

|  | FoldX (Stability) | FoldX (SASA) |
|---|---|---|
| Random mut. | $2.79 \pm 0.22$ | $12550.26 \pm 56.34$ |
| LaMBO-2 | $3.19 \pm 0.58$ | $12456.10 \pm 126.64$ |
| CbAS | $3.65 \pm 0.23$ | $12376.65 \pm 298.30$ |
| VSD | $\mathbf{4.20 \pm 0.42}$ | $12537.97 \pm 186.35$ |
| GenBO | $3.28 \pm 0.35$ | $\mathbf{13285.42 \pm 221.60}$ |

Table 5: FoldX average maximum outcome for the best-performing variant of each method

| Method | Average Runtime |
|---|---|
| CbAS | $53.38 \text{ s} \pm 2.05 \text{ s}$ |
| VSD | $42.89 \text{ s} \pm 2.56 \text{ s}$ |
| GenBO | $\mathbf{14.88 \text{ s} \pm 0.26 \text{ s}}$ |

Table 6: Average run times with standard deviations for the ALOHA text optimization problem.

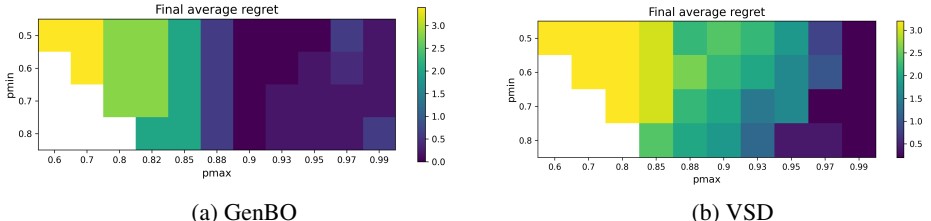

(a) GenBO          (b) VSD

Figure 5: Final average simple regret (the lower the better) for GenBO and VSD as a function of the minimum and maximum percentile in the annealing schedule for the text optimization problem.

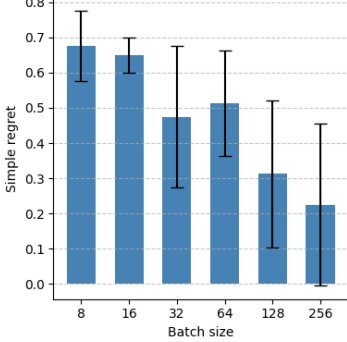

Figure 6: Batch evaluation size $B$ ablation on Ehrlich benchmark of length 32. The plot presents the final average simple regret for each $B$ setting, with error bars corresponding to $\pm 1$ standard deviation. All variants were run for the same number of BO rounds as in the original experiment.

