# OpenReview forum: "Generative Bayesian Optimization: Generative Models as Acquisition Functions"
_ICLR.cc/2026/Conference — ICLR 2026 Poster_

### Official Review · Reviewer_Vkoy · 2025-10-29

**Soundness:** 3
**Presentation:** 2
**Contribution:** 3
**Rating:** 6
**Confidence:** 3

**Summary:**

This paper tackles the challenge of how to approach Batch based Bayesian optimisation. The challenge in Batch BO is how to query a set of evaluation locations as we want them to be diverse. The traditional approach turns this into a sequential decision problem. The idea in this paper is to learn a separate model that parametrises the support of the acquisition function such that this can be used to sample candidate locations. The paper presents several different approaches on how such a model can be defined both using the acquisition function perspective and one of preference losses.

**Strengths:**

The paper is generally well written and honest in what it proposes not overstating what is proposed, thank you for this. The work builds significantly on the work proposed in [1] but proposes new and novel mechanisms to train the model and the use-case here is directly BO which was only suggested in the Appendix of [1]. The preference based training is to my knowledge novel in this modelling scenario. The paper tackles a very important problem in any sequential decision making system where multiple proposals can be exploited at the same time.

[1] Stein berg, D. M., Oliveira, R., Ong, C. S., & Bonilla, E. V. (2024). Variational Search Distributions. CoRR, (), .

**Weaknesses:**

While I appreciate the extensive background writing it somewhat also pollutes the story a bit. For example, I am not actually sure why BO using latent representations is part of the Batch-BO introduction. While it is an important class of models it is in somewhat not used to "paint a picture" of what you are proposing. I think the work would come into a better context if some of the background was shortened and details where only introduced if they helped the story of the paper. This might be completely personal opinion but I find the text a bit scattered up to Section 3 and some of this could be moved to after the proposed method. Again just a personal opinion that has had no impact on my score or review.

One thing that I didn't fully understand is what is the specific q that you are using and what the p_0 is for the experiments that requires this. Maybe I've missed it in the paper but it would make the results easier to interpret.

**Questions:**

Could you expand on what the specific model for q is and detail what p_0 is in the experimental settings that requires this?

---

> ### Author Response · Authors · 2025-11-21
> **Thank you**
>
> We would like to thank the reviewer for their positive feedback and helpful insights. We address your concerns and specific question below.
>
> > While I appreciate the extensive background writing it somewhat also pollutes the story a bit. For example, I am not actually sure why BO using latent representations is part of the Batch-BO introduction. While it is an important class of models it is in somewhat not used to "paint a picture" of what you are proposing. I think the work would come into a better context if some of the background was shortened and details where only introduced if they helped the story of the paper. This might be completely personal opinion but I find the text a bit scattered up to Section 3 and some of this could be moved to after the proposed method. Again just a personal opinion that has had no impact on my score or review.
>
> We thank the reviewer for pointing this out and acknowledge that the background section turned out somewhat extensive. Our idea was to merge related work and technical background together, due to the need to introduce some technical details first before discussing some of the literature. Thus, we will incorporate the reviewer feedback in the  revised version, by shrinking the background and moving some of it to a dedicated related work section.
>
> > Could you expand on what the specific model for $q$ is and detail what $p_0$ is in the experimental settings that requires this?
>
> We used mean-field categorical (for  low-dimensional problems) and transformer models (more generally) for both the proposal $q$ and the prior $p_0$ (i.e., a model trained with an initial randomized dataset $\mathcal{D}_0$ at the beginning of the process). In particular, these were the same generative models used by VSD and CbAS for each problem. Our mention of that can be found in sections 5.1 and 5.2 (under *Experiments*), where we state that GenBO, VSD and CbAS share the same generative backbone. The details are specified in Appendix C.

---

> > ### Author Response · Authors · 2025-11-30
> >
> > We believe we have now addressed reviewer Vkoy's concerns in the latest revision of the paper:
> > - We have split the extensive background section into a more highly focused background section, _and a new related work section_ that highlights the difference between GenBO and other generative BO-style methods.
> > - We have expanded the in-text descriptions of the experimental settings, including model architectures, and we also give a more thorough interpretation of the results (with additional figures).

---

### Official Review · Reviewer_szDh · 2025-10-30

**Soundness:** 3
**Presentation:** 3
**Contribution:** 3
**Rating:** 6
**Confidence:** 4

**Summary:**

This paper proposes a framework called generative Bayesian optimization (BO), which transforms generative models into candidate samplers for batch BO. By leveraging ideas from direct preference optimization (DPO) in RLHF, the approach trains generative models directly using noisy utility values derived from observations, making the proposal distribution's density proportional to the acquisition function. This eliminates the need for intermediate surrogate models, reducing computational cost and approximation errors compared to prior methods like VSD or BORE. The framework supports two training strategies: a DPO-like loss for preference-based feedback and divergence minimization with utility-weighted samples for general rewards. It is particularly suited for large-batch scaling, high-dimensional spaces, non-continuous designs, and combinatorial problems. Theoretically, the authors show that the sequence of generative models asymptotically concentrates at the global optima under certain conditions. Empirical evaluations on challenging high-dimensional optimization tasks demonstrate improved performance for large batches.

**Strengths:**

1. The key innovation is treating generative models as direct acquisition functions via DPO-inspired training, bypassing the two-stage process (surrogate then generative) in existing methods. This simplifies the pipeline, makes it more general to utility types, and enables efficient sampling for large batches in complex spaces. The extension to batch BO addresses a practical need in parallel evaluations, like simulations or experimental design.
2. The paper provides convergence analysis, showing asymptotic concentration of proposal distributions around optima. Assumptions seem reasonable (e.g., bounded domain, sub-Gaussian noise, smooth objectives), building on standard BO theory (e.g., Bull, 2011 for EI; Srinivas et al., 2010 for regret).
3. Scalable to high dimensions and combinatorial domains, where traditional GP-based BO struggles. Compatibility with pre-trained generative models allows fine-tuning for optimization, with low sampling cost. The framework generalizes beyond preferences to arbitrary utilities, broadening applicability to areas like hyperparameter tuning, materials, and robotics.
4. Equations are well-explained, and the motivation from real-world batch scenarios is compelling.

**Weaknesses:**

1. While claiming reduced cost by avoiding surrogates, training generative models (especially large ones like LLMs) on utilities could still be expensive in practice, particularly for online BO with frequent retraining. No discussion of training overhead vs. baselines.
2. Focuses on batch BO, but less emphasis on sequential cases or comparison to non-generative batch method. Robustness to noisy utilities or misspecified priors isn't addressed. For combinatorial spaces, discretization or embedding strategies aren't detailed.

**Questions:**

Please see the Weaknesses.

---

> ### Author Response · Authors · 2025-11-21
> **Response**
>
> We thank the reviewer for their constructive feedback and insights.
>
> > 1. While claiming reduced cost by avoiding surrogates, training generative models (especially large ones like LLMs) on utilities could still be expensive in practice, particularly for online BO with frequent retraining. No discussion of training overhead vs. baselines.
>
> One of the main advantages in GenBO is that costs scale linearly, instead of cubically, with the number of data points. Hence, despite the need for retraining, we are still proposing an approach that is more computationally scalable than GP-based BO, especially in higher-dimensional settings with large batches. However, one way of saving computational costs would be by applying low-rank approximations, like LoRA, to retrain the models. As the models converge to the target distribution, it is to be expected that the amount of change in their parameters from round to round should become relatively small. Hence, one could warm start each round $t$ with the optimized parameters from the previous round $t-1$ and then run LoRA to provide low-rank updates for the model parameters. Nevertheless, we highlight that, for this paper, we only considered relatively small models for protein sequence generation, which do not require the level of computational resources usually consumed by LLMs. We have clarified this in the new literature review section.
>
> With regards to runtime, in the FoldX problem, which involves realistic, complex numerical simulations, we observed an average of 40% reduction in runtime relative to the closest approach and next-best-performing baseline, VSD. As we used the same generative models that VSD used, the difference in runtime is therefore due to the absence of a secondary surrogate model for the expected utilities (i.e., the class probability estimator that VSD uses). _We have included runtime comparisons in the revision_ on the ALOHA experiment, see table 6 in the appendix.
>
> > 2. Focuses on batch BO, but less emphasis on sequential cases or comparison to non-generative batch method. Robustness to noisy utilities or misspecified priors isn't addressed. For combinatorial spaces, discretization or embedding strategies aren't detailed.
>
> **Focus of the paper.** We limited the scope of this paper to batch-sequential problems in large combinatorial spaces with relatively large batch sizes, which non-generative approaches typically do not scale to, due to the rapidly increasing costs of conditional batch generation with typical BO models, like Gaussian processes. Even in the ALOHA problem, where the batch size was small ($B = 8$), the combinatorial nature problem leads to a search space of $26^5 \approx 11.8$ million discrete candidates, which we cannot easily enumerate for selecting batch candidates with typical algorithms found in popular BO libraries, like BoTorch. One then needs to, instead, consider other approaches, such as latent-space BO, which LaMBO-2 is a representative of, or generative methods, like CbAS, VSD and GenBO.
>
> **Noisy utilities.** Utilities are assumed to be noisy by default. Note that we model expected utilities, since the observations are noisy as well. Recall that utilities are a direct function of the observation values, e.g., $u(y) = \mathbb{I}[y \geq \tau]$ for PI.
>
> **Embeddings.** We would also like to emphasize that our approach does not involve latent-space-based optimization. We are sampling sequences directly with our models, which are detailed in Appendix C. _This point has been clarified in the new literature review section_. For the VSD and CbAS baselines, which use a class probability estimator (CPE), i.e., a classifier, we detail in Figure 3 how we embed the tokens (amino acids are directly tokenized). For the generative models, we just treat amino-acids as tokens directly, and so each amino acid is generated from a categorical distribution, similar to Algorithm 14 in Phuong and Hutter (2022).
>
> **Misspecification.** Regarding prior misspecification, as a first step, we had to focus on the well-specified case to formalize the theoretical framework, carry on its analysis and derive guarantees. In future work, we plan to investigate the case of misspecified priors. One possible avenue is to replace the KL divergence in our formulations by another divergence which is more robust to misspecification, such as Renyi divergences with their $\alpha$ parameter that can be adjusted according to the potential level of misspecification. This is the general idea behind generalized all inference frameworks. We will add this to the limitations and future directions section of the paper.
>
> ### References
> * Knoblauch, J., Jewson, J., \& Damoulas, T. (2022). An optimization-centric view on Bayes' rule: Reviewing and generalizing variational inference. *Journal of Machine Learning Research, 23*(132), 1-109.
> * Phuong, M., \& Hutter, M. (2022). Formal algorithms for transformers. *arXiv preprint arXiv:2207.09238*.

---

### Official Review · Reviewer_REEK · 2025-11-01

**Soundness:** 3
**Presentation:** 4
**Contribution:** 3
**Rating:** 6
**Confidence:** 4

**Summary:**

This paper presents the direct use of generative models as samplers for design selection in batched Bayesian optimization. Unlike prior two-stage approaches that first fit a surrogate or regressor and then train a generator, the authors propose a single-stage method that learns a generative model whose density is proportional to the expected utility. Two approaches are introduced. The first is preference-based learning, analogous to Direct Preference Optimization (DPO), using a contrastive objective built from pairwise utilities, with a robust variant that accounts for noisy labels and sign flips. The second is a KL-divergence route that trains the generator to match the utility-weighted target distribution via a balanced forward KL, implemented with importance-weighted sampling to yield an unbiased objective, along with an alternative KL derived from Bregman divergences to avoid allocating high probability mass to low-utility regions. The paper also provides theoretical analysis showing convergence to the utility-based target distribution and concentration of the generative BO proposals at the true optimum.

**Strengths:**

1. The core idea of a single-model training approach for the generative sampler is well motivated and clearly presented.

2. The method attempts to generalize across expected-utility acquisition function variants.

3. Scalability is linked to batched BO, making the approach practically relevant if successful.

4. The paper provides theoretical analyses that support the claims, with careful attention to both convergence and optimality.

5. The empirical evaluation compares the proposed methods against relevant baselines.

**Weaknesses:**

1. The empirical analysis is well motivated but does not make a fully convincing case for the proposed algorithm.

2. Related to the above, the results do not show consistent superiority over all baselines on the more complex problems. While the authors do note cases of underperformance with provided justifications, it would help to comment on how the approach could be improved to close those gaps.

3. The plots are quite difficult to read.

3. Minor: while the paper claims generalization to utility-based acquisitions, it does not clearly discuss extensions to other acquisition families such as GP-UCB or Thompson Sampling.

**Questions:**

1. I am willing to raise my score if the authors can address my main concerns about the empirical results, specifically, by demonstrating stronger robustness and clearer superiority over baselines where possible.

2. While the framework targets utility-based acquisitions, can you comment on extensions to other acquisition families such as Thompson Sampling and GP-UCB? Perhaps such extensions are not straightforward or not possible? An explanation would be great.

---

> ### Author Response · Authors · 2025-11-21
> **Response**
>
> We would like to thank the reviewer for their constructive feedback and careful consideration of our work. Our revised PDF addresses the concerns regarding readability and presentation issues. Other concerns are addressed below.
>
> > I am willing to raise my score if the authors can address my main concerns about the empirical results, specifically, by demonstrating stronger robustness and clearer superiority over baselines where possible.
>
> We would like to highlight that we outperform or reach similar performance to the baselines in most experiments, except for the FoldX stability problem (Fig. 1b), and we do so using only a single generative model, without the need for a class probability estimator (as in VSD and CbAS), deep ensembles (as in LaMBO-2) or Gaussian process (as in traditional BO). Even if matching performance, we still save on computational costs and have an advantage in simplicity of real-world deployments (one-fewer critical component to architect and train). For the FoldX problem, for instance, which involves realistic numerical simulations, we observed a 40\% reduction in runtime when compared to the next best performing method, VSD. In addition, *in our revision, we have managed to outperform all other baselines on the Ehrlich problem with sequence length 32* (Fig. 2b). The key to that was an exponential regularization term. As described in our theoretical results (Lemma 2), the typical L2 regularization can be seen as a particular instance of a more general RKHS-based regularization framework, corresponding to an assumption of linearity in the model's parameter space (see Remark 2 in the Appendix), which is somewhat restrictive. A more general approach is to assume an exponential dot-product kernel to define the RKHS of functions over the model's parameter space. The exponential dot-product is known to be a universal kernel over the space of continuous functions on a given (compact) domain, leading to better representation power. Such regularization was key to achieving better results on the Ehrlich-64 problem, and now also on the Ehrlich-32 problem. We will add the new results to the revised paper.
>
> > While the framework targets utility-based acquisitions, can you comment on extensions to other acquisition families such as Thompson Sampling and GP-UCB? Perhaps such extensions are not straightforward or not possible? An explanation would be great.
>
> GP-UCB and Thompson sampling are not trivially expressible as expected utilities in terms of only the observations, but potentially our framework could be extended to other functionals (i.e., beyond expectations) of the utilities distribution, similar to the ideas in this recent paper by De Santi et al. (2025, below) and other frameworks extending generalized variational inference to different risk measures. We will remark on this point in the paper as a limitation and possible direction of future work.
>
> ### References
> * De Santi, R., Vlastelica, M., Hsieh, Y. P., Shen, Z., He, N., \& Krause, A. (2025). Flow Density Control: Generative Optimization Beyond Entropy-Regularized Fine-Tuning. In *The Exploration in AI Today Workshop at ICML 2025*.

---

### Official Review · Reviewer_Jdpb · 2025-11-01

**Soundness:** 2
**Presentation:** 2
**Contribution:** 1
**Rating:** 2
**Confidence:** 4

**Summary:**

This paper introduces Generative Bayesian Optimization (GenBO), a framework where a generative model is used directly as the acquisition function for Bayesian Optimization (BO).
Traditional generative BO methods often use a two-stage process: first, training a surrogate model (like a Gaussian Process), and second, training a generative model based on the surrogate's predictions. This can lead to compounding errors.
GenBO, inspired by Direct Preference Optimization (DPO), bypasses the intermediate surrogate model. It trains the generative model directly on simple, observed utility values (such as Expected Improvement or Probability of Improvement). The resulting model's probability density becomes proportional to the expected utility.
Therefore, sampling from this trained generative model is equivalent to sampling candidates from the acquisition function, simplifying the pipeline, reducing approximation errors, and enabling scalability to large batches and complex, high-dimensional spaces like protein design.

**Strengths:**

The author provided the theroetical analysis of their proposed method.

**Weaknesses:**

**Illegible Figures**

The figures are poorly rendered and extremely difficult to interpret. The hues selected for the proposed method and the various baselines are too similar, making the different plots (e.g., performance curves) indistinguishable. Compounding this issue, the legend is rendered at a font size that is far too small to be legible. As a result, it is impossible for the reviewer to accurately interpret the experimental results or validate the paper's claims.

**Insufficient Experimentation and Analysis**

The empirical validation is severely lacking. The paper presents minimal experiments with what appears to be a complete absence of analysis. The claims are not adequately supported by empirical evidence.

**Trivial Methodology and Limited Contribution**

The core idea seems to be training an LLM on (data, utility function value) pairs via DPO to generate better data points. This appears to be a trivial application of DPO to the Bayesian Optimization (BO) setting. If one were tasked to "use DPO for BO," this is the most straightforward idea one would likely conceive. The conceptual novelty and overall contribution seem too limited.

**Clarity on Training Data**

The amount of data used for training the LLM is not specified. BO inherently assumes an expensive-to-evaluate setting, which typically yields sparse data. It is highly questionable whether this data regime is sufficient to fine-tune an LLM effectively.

**Baseline selection**

The field of generative model-based BO is well-established, often referred to as LBO (Latent Space Bayesian Optimization)[1~7]. The paper fails to position itself relative to this body of work. What are the specific advantages of this DPO-based approach compared to existing LBO methods? The lack of comparison makes it impossible to assess the method's practical or theoretical benefits.

**Reference**

[1] Tripp, Austin, Erik Daxberger, and José Miguel Hernández-Lobato. "Sample-efficient optimization in the latent space of deep generative models via weighted retraining." Advances in Neural Information Processing Systems 33 (2020): 11259-11272.

[2] Maus, Natalie, et al. "Local latent space bayesian optimization over structured inputs." Advances in neural information processing systems 35 (2022): 34505-34518.

[3] Lee, Seunghun, et al. "Advancing bayesian optimization via learning correlated latent space." Advances in Neural Information Processing Systems 36 (2023): 48906-48917.

[4] Chu, Jaewon, et al. "Inversion-based latent bayesian optimization." Advances in Neural Information Processing Systems 37 (2024): 68258-68286.

[5] Moss, Henry B., Sebastian W. Ober, and Tom Diethe. "Return of the latent space COWBOYS: Re-thinking the use of VAEs for Bayesian optimisation of structured spaces." arXiv preprint arXiv:2507.03910 (2025).

[6] Grosnit, Antoine, et al. "High-dimensional Bayesian optimisation with variational autoencoders and deep metric learning." arXiv preprint arXiv:2106.03609 (2021).

[7] Lee, Seunghun, et al. "Latent bayesian optimization via autoregressive normalizing flows." arXiv preprint arXiv:2504.14889 (2025).

**Questions:**

See the weakness section above.

---

> ### Author Response · Authors · 2025-11-21
> **Response - Part I**
>
> We would like to thank the reviewer for their detailed feedback and comments. Regarding presentation issues, especially concerning the experimental results, these have been addressed in our revision, whose PDF will be uploaded here. We address the reviewer's remaining concerns below.
>
> > **Insufficient Experimentation and Analysis.** The empirical validation is severely lacking. The paper presents minimal experiments with what appears to be a complete absence of analysis. The claims are not adequately supported by empirical evidence.
>
> We present four separate experiments in this work, one of which has three different configurations. One of these (Aloha) is a simple proof-of-concept experiment, whereas the remaining are *challenging high dimensional sequence* design tasks. In particular, the FoldX-based design tasks (Stability and SASA) are using an industry standard in-silico protein simulator, which is commonly used to screen protein designs before wet-lab testing. As such we contest the point that the empirical validation is severely lacking.
>
> We do acknowledge the lack of a detailed discussion for some of the results in the experiments section. In the revised version, we expanded the discussion of experimental results, and have improved the rendering of the figures.
>
> > **Trivial Methodology and Limited Contribution.**
> The core idea seems to be training an LLM on (data, utility function value) pairs via DPO to generate better data points. This appears to be a trivial application of DPO to the Bayesian Optimization (BO) setting. If one were tasked to "use DPO for BO," this is the most straightforward idea one would likely conceive. The conceptual novelty and overall contribution seem too limited.
>
> We would like to clarify that there is no LLM in our paper. We propose a general framework for Bayesian optimization using generative models that avoids the construction of surrogate models (e.g., a Gaussian process or a probabilistic classifier, etc.). One of the objective functions in our framework is inspired by DPO, *but* it is not the only loss function we are proposing in the paper, as we also consider KL-based losses. In fact, other than the ALOHA Ngram problem, DPO and robust DPO were outperformed by KL-based loss functions in higher-dimensional optimization problems. In addition, we see the relative simplicity of the framework we are proposing as one of its strengths. The main idea of this paper was indeed to simplify the setting of generative black-box optimization, which would often involve two probabilistic models to optimize (i.e., an objective function surrogate and a generative model), to instead rely on a single generative model, cutting down in half the need for computational resources, including memory and runtime. We have also derived *theoretical results* showing that one can essentially learn any target probability distribution in these settings with non-i.i.d. and heavily dependent data under mild assumptions, guarantees which are generally lacking in the generative black-box optimization literature. In short, we are the first (to our knowledge) to prove that DPO and other reward-model-free divergence-based losses can *even be used* to solve black-box optimization problems, which we believe is a substantial contribution to the literature and will significantly simplify the real-world application of generative black-box optimization. We will clarify this contribution in our paper.

---

> > ### Author Response · Authors · 2025-11-21
> > **Response - Part II**
> >
> > > **Clarity on Training Data.** The amount of data used for training the LLM is not specified. BO inherently assumes an expensive-to-evaluate setting, which typically yields sparse data. It is highly questionable whether this data regime is sufficient to fine-tune an LLM effectively.
> >
> > We would like to clarify once again  that there are not large language models (LLMs) in our paper. Our experiments were run using categorical mean-field models and causal transformers for protein sequences. The amount of data used to train the models consists of the batches accumulated across iterations, i.e., at iteration/round $t \in \{1, \dots, T\}$ (between 10 and 32, depending on the problem), we have $Bt$ data points, where $B$ corresponds to the batch size (ranging from 8 to 128 depending on the problem), which in our case corresponds to the number of generated candidates evaluated (in parallel) at each round, hence, not a stochastic gradient descent mini-batch size. For our models, the amount of data aggregated was sufficient to generate useful candidates per round.
> >
> > We would also like to comment that, in contrast to a prediction-oriented framework, in BO we are not necessarily aiming at "perfect" predictions, which would make the models overly exploitative and require as much data as possible. A small amount of data in each round is not a drawback, as other mechanisms, such as regularization and random initialization of the model parameters, take over and allow the model to produce more diverse samples, promoting exploration. Once enough data is accumulated, these mechanisms naturally decay in favor of a more exploitative approach to produce samples closer to the optimum. These processes have been well studied in the case of Gaussian processes, but we have observed similar behavior in the generative BO setting. We will expand on this discussion in the paper.
> >
> > >**Baseline selection.** The field of generative model-based BO is well-established, often referred to as LBO (Latent Space Bayesian Optimization)[1--7]. The paper fails to position itself relative to this body of work. What are the specific advantages of this DPO-based approach compared to existing LBO methods? The lack of comparison makes it impossible to assess the method's practical or theoretical benefits.
> >
> > We appreciate the list of references recommended by the reviewer. However, we would like to emphasize that we have discussed a few works on latent-space BO in our background (including related work) section under the "Batch BO" paragraph. In addition, our experimental results have included comparisons against LaMBO-2 (Gruver et al., 2023), which is the successor of Latent-space Multi-objective Bayesian Optimization (LaMBO) by Stanton et al. (2022), an LBO method. Most other generative model-based BO approaches, like LBO, require an explicit surrogate for the objective function, besides the generative model, doubling computational costs. Our work requires only a single model (i.e., the generative model), instead. We would also like to remark that LBO is not the only genre of generative BO, with methods such as DbAS and CbAS and other estimation of distribution/cross entropy methods and evolutionary strategies making up another large body of literature. The literature review section of Steinberg et al. (2025) provides an overview of how many of these methods relate.

---

> ### Comment · Reviewer_Jdpb · 2025-11-26
>
> Thank you to the authors for their kind and detailed response.
>
> **Insufficient Experimentation and Analysis**
>
> I would like to clarify that my comment regarding "insufficient experimentation" was not referring to the number of optimization tasks. Rather, I intended to point out the lack of experimental analysis regarding the proposed methodology, which is standard in most research papers (e.g., ablation studies or behavioral analysis beyond just optimization performance). Additionally, Figures 1 and 2 still require significant revision. In Figure 1(b), the legend is too small to be legible. In Figure 1(c), the legend obstructs the optimization curves, making it impossible to distinguish the results. In Figure 2, all experimental results are depicted using a similar color palette, making it difficult to differentiate between them.
>
> **Trivial Methodology and Limited Contribution**
>
> First, I apologize for the confusion caused by my mention of LLMs. However, I still harbor fundamental doubts regarding this methodology. Research on BO that trains generative models to preferentially produce data with higher objective function values already exists.[1,2] The primary difference between those works and this paper is the presence or absence of a surrogate model. Existing studies likely utilized surrogate models to achieve higher performance, even though they also trained generative models to target high function values. Therefore, I remain skeptical as to whether simply removing the surrogate model constitutes a significant contribution.
>
> **Baseline Selection**
>
> LaMBO-2 is a strong model; however, it was published in 2023 and is arguably becoming slightly outdated. Since then, several studies on BO using generative models with superior optimization performance have been proposed. While I acknowledge that LBO is not the only genre of Generative BO, I do not believe this is a valid justification for omitting comparisons with the latest state-of-the-art LBO research.
>
>
> Additionally, upon reviewing the comments from other reviewers, it appears that I may be the outlier in not fully appreciating the value of this paper. As a researcher deeply interested in BO, I will strive to understand the merit of this work, but as of now, I have not yet been convinced of the distinct advantages this paper offers.
>
> **Reference**
>
> [1] Tripp, Austin, Erik Daxberger, and José Miguel Hernández-Lobato. "Sample-efficient optimization in the latent space of deep generative models via weighted retraining." Advances in Neural Information Processing Systems 33 (2020): 11259-11272.
>
> [2] Lee, Seunghun, et al. "Advancing bayesian optimization via learning correlated latent space." Advances in Neural Information Processing Systems 36 (2023): 48906-48917.

---

> > ### Author Response · Authors · 2025-11-30
> >
> > We thank reviewer Jdpb for following up on our response, and clarifying their concerns. We realise they have now been blocked from responding, and we hope this response can clear up any remaining issues for the AC.
> >
> > > **Insufficient Experimentation and Analysis**
> >
> > We have added some ablation studies to the appendix, as well as additional results.
> > - We present batch diversity results for the Stability and SASA experiments in figure 3.
> > - We study the effects of the acquisition function thresholding schemes on GenBO and VSD, as well as the effect of batch size on GenBO (figures 5 and 6).
> > - We give ALOHA experimental timings in table 6.
> > - We clarify final round performance for all methods in tables 4 and 5.
> >
> > To address the legibility concern raised, we have replotted all results improving the legibility of the axes and legends, and have increased the dynamic range of the color palette.
> >
> > > **Trivial Methodology and Limited Contribution**
> >
> > Unfortunately we believe reviewer Jdpb conceptually misunderstands our work and this is the source of their low score.
> >
> > We respectfully clarify that our method is not a minor variant of latent-space Bayesian optimisation (LBO). The cited works [1,2] and other LBO methods operate in a learned continuous latent manifold, $\mathcal{Z}$, requiring an encoder ($\mathcal{X} \to \mathcal{Z}$), decoder ($\mathcal{Z} \to \mathcal{X}$), and surrogate ($\mathcal{Z} \to \mathcal{Y}$). BO is then performed on this manifold, which must remain either fixed after pre-training, potentially limiting the exploitation of under-sampled regions of the original design space ([Lee et al. 2025] uses normalising flows to help mitigate this issue), or has to be carefully retrained as to not invalidate current progress of the surrogate [1].
> >
> > Our method is a direct-space generative optimiser: it requires _no latent manifold_ and _no surrogate_, and instead reweights a generative model directly over discrete sequences, $\mathcal{X}$. This removes the core LBO failure mode (manifold mismatch), simplifies the pipeline (components can be validated independently), and yields qualitatively different optimisation behaviour -- closer to recent direct-space frameworks such as VSD. Thus, the contribution is not “removing the surrogate,” but replacing the latent-BO paradigm with a more simple, robust and appropriate design-space method, _and then providing rigorous convergence guarantees_.
> >
> > We have added these points to the new literature review section in our paper to help clarify any misconceptions.
> >
> > > **Baseline Selection**
> >
> > Though LaMBO-2 was published in 2023, it is still considered SOTA (along with VSD, 2025) for the type of long-sequence protein online-optimization problems we consider in this work. All of the other more recent LBO methods suggested by the reviewer have only been applied to small molecule tasks, and the generative models they use (VAE, NF, etc.) are known to be less effective for proteins, and so we believe they would be at a disadvantage in a direct comparison with our method and chosen baselines.

---

### Author Response · Authors · 2025-11-29
**Revised PDF**

Dear Reviewers and AC,

We have uploaded a revised version of our paper addressing the issues raised by the reviewers. The following table summarises our main changes in response to each issue with referral to the corresponding reviewer.

| Issue | Revision |
|-------------------------|----------|
| Excessive background (Vkoy) | We split background and related work into separate sections (Sec. 2 and 5, respectively) to improve readability and flow of the paper's narrative. |
| Related work (Jdpb) | We have expanded our discussion on related work on latent-space BO methods and also added a paragraph about work on using LLMs for BO, contrasting them with the more general setting in our paper, which does not focus on language models. |
| Performance robustness (REEK) | We have obtained new results on the text optimization and Ehrlich (with sequence length 32) problems, surpassing all baselines. |
| Illegible figures (Jdpb and REEK)        |   Figures have been replotted with larger fonts and more distinguishable colors.  In addition, we have summarized all experiment results in numerical tables (Table 4 and 5) to ensure legibility across all tasks. |
| Insufficient empirical analysis (Jdpb)   | We have significantly expanded the analysis in the experiments section (now Sec. 6), including ablations and other auxiliary results to aid in the performance analysis, discussing potential reasons for the observations.  |
| Trivial contribution (Jdpb)   | We believe that reviewer Jdpb conceptually misunderstands our work. We have attempted to clarify that our work is separate from latent-space BO (LBO) methods, and side-steps many of their issues by directly operating in the original design space. Furthermore, our method does not use an LLM, but task-specific generative models. We believe we have addressed this in the discussion below, and we have clarified the distinction between our method (and other direct design space methods) from LBO methods in the paper's new related work section. We have also clarified in the revised paper that our method does not make use of LLMs. |
| Computational costs (szDh) | We have added a table with runtime (Table 6) measured on the text optimization (ALOHA) problem comparing GenBO against baselines. |
| Experiment details | We include additional details on experiment settings in Appendix D. |

---

### Author Response · Authors · 2025-11-30
**Thank you to the reviewers**

As a final note to the reviewers: we thank you all for spending the time in preparing high quality, constructive reviews of our work. We regret not being able to engage with you further, and we sincerely believe that your input so far has led to a much higher quality revision of our paper.

All the best,

Authors.

---

### Meta-Review · Area_Chair_VyMh · 2026-01-09

**Summary:**

The outlier reviewer with negative comments had two concerns:

- ""Methodology""
- Experimental results (baselines, especially in latent space BO) and problems considered.

**Reviewer Concerns:**

As an AC, I generally don't put much stock in "methodological novelty" complaints as a whole, as rejecting papers on that basis tends to result in over optimization towards complexity rather than novelty. I think the method is overall good.

I do think some of the experimental criticisms are valid, and while I'm definitely going to recommend acceptance I do want to highlight:

> All of the other more recent LBO methods suggested by the reviewer have only been applied to small molecule tasks, and the generative models they use (VAE, NF, etc.) are known to be less effective for proteins.

5 minutes with a search engine turns up latent space BO results on proteins, peptides, and even DNA (where sequences are going to be vastly longer than considered here). You could probably ask for those VAEs?

>  And so we believe they would be at a disadvantage in a direct comparison with our method and chosen baselines.

If only there was some way of finding that out ;-).

**Reviewer Scores:**

I think it's a good paper, and most of the reviewers did too.

---

### Decision · Program_Chairs · 2026-01-26

Accept (Poster)